# Exploration in Structured Reinforcement Learning

**Jungseul Ok**
KTH, EECS
Stockholm, Sweden
ockjs@illinois.edu

**Alexandre Proutiere**
KTH, EECS
Stockholm, Sweden
alepro@kth.se

**Damianos Tranos**
KTH, EECS
Stockholm, Sweden
tranos@kth.se

## Abstract

We address reinforcement learning problems with finite state and action spaces where the underlying MDP has some known structure that could be potentially exploited to minimize the exploration rates of suboptimal (state, action) pairs. For any arbitrary structure, we derive problem-specific regret lower bounds satisfied by any learning algorithm. These lower bounds are made explicit for unstructured MDPs and for those whose transition probabilities and average reward functions are Lipschitz continuous w.r.t. the state and action. For Lipschitz MDPs, the bounds are shown not to scale with the sizes $S$ and $A$ of the state and action spaces, i.e., they are smaller than $c \log T$ where $T$ is the time horizon and the constant $c$ only depends on the Lipschitz structure, the span of the bias function, and the minimal action sub-optimality gap. This contrasts with unstructured MDPs where the regret lower bound typically scales as $SA \log T$. We devise DEL (Directed Exploration Learning), an algorithm that matches our regret lower bounds. We further simplify the algorithm for Lipschitz MDPs, and show that the simplified version is still able to efficiently exploit the structure.

## 1 Introduction

Real-world Reinforcement Learning (RL) problems often concern dynamical systems with *large* state and action spaces, which make the design of efficient algorithms extremely challenging. This difficulty is well illustrated by the known *regret* fundamental limits. The regret compares the accumulated reward of an optimal policy (aware of the system dynamics and reward function) to that of the algorithm considered, and it quantifies the loss incurred by the need of exploring sub-optimal (state, action) pairs to learn the system dynamics and rewards. In online RL problems with undiscounted reward, regret lower bounds typically scale as $SA \log T$ or $\sqrt{SAT}$[1], where $S$, $A$, and $T$ denote the sizes of the state and action spaces and the time horizon, respectively. Hence, with large state and action spaces, it is essential to identify and exploit any possible structure existing in the system dynamics and reward function so as to minimize exploration phases and in turn reduce regret to reasonable values. Modern RL algorithms actually implicitly impose some structural properties either in the model parameters (transition probabilities and reward function, see e.g. [Ortner and Ryabko, 2012]) or directly in the $Q$-function (for discounted RL problems, see e.g. [Mnih et al., 2015]). Despite the successes of these recent algorithms, our understanding of structured RL problems remains limited.

In this paper, we explore structured RL problems with finite state and action spaces. We first derive problem-specific regret lower bounds satisfied by any algorithm for RL problems with any arbitrary structure. These lower bounds are instrumental to devise algorithms optimally balancing exploration and exploitation, i.e., achieving the regret fundamental limits. A similar approach has

been recently applied with success to stochastic bandit problems, where the average reward of arms exhibits structural properties, e.g. unimodality [Combes and Proutiere, 2014], Lipschitz continuity [Magureanu et al., 2014], or more general properties [Combes et al., 2017]. Extending these results to RL problems is highly non trivial, and to our knowledge, this paper is the first to provide problem-specific regret lower bounds for structured RL problems. Although the results presented here concern ergodic RL problems with undiscounted reward, they could be easily generalized to discounted problems (under an appropriate definition of regret).

Our contributions are as follows:

1. For ergodic structured RL problems, we derive problem-specific regret lower bounds. The latter are valid for any structure (but are structure-specific), and for unknown system dynamics and reward function.

2. We analyze the lower bounds for unstructured MDPs, and show that they scale at most as $\frac{(H+1)^2}{\delta_{\min}} SA \log T$, where $H$ and $\delta_{\min}$ represent the span of the bias function and the minimal state-action sub-optimality gap, respectively. These results extend previously known regret lower bounds derived in the seminal paper [Burnetas and Katehakis, 1997] to the case where the reward function is unknown.

3. We further study the regret lower bounds in the case of Lipschitz MDPs. Interestingly, these bounds are shown to scale at most as $\frac{(H+1)^3}{\delta_{\min}^2} S_{\text{lip}} A_{\text{lip}} \log T$ where $S_{\text{lip}}$ and $A_{\text{lip}}$ only depend on the Lipschitz properties of the transition probabilities and reward function. This indicates that when $H$ and $\delta_{\min}$ do not scale with the sizes of the state and action spaces, we can hope for a regret growing logarithmically with the time horizon, and independent of $S$ and $A$.

4. We propose DEL, an algorithm that achieves our regret fundamental limits for any structured MDP. DEL is rather complex to implement since it requires in each round to solve an optimization problem similar to that providing the regret lower bounds. Fortunately, we were able to devise simplified versions of DEL, with regret scaling at most as $\frac{(H+1)^2}{\delta_{\min}} SA \log T$ and $\frac{(H+1)^3}{\delta_{\min}^2} S_{\text{lip}} A_{\text{lip}} \log T$ for unstructured and Lipschitz MDPs, respectively. In absence of structure, DEL, in its simplified version, does not require to compute action indexes as done in OLP [Tewari and Bartlett, 2008], and yet achieves similar regret guarantees without the knowledge of the reward function. DEL, simplified for Lipschitz MDPs, only needs, in each step, to compute the optimal policy of the estimated MDP, as well as to solve a simple linear program.

5. Preliminary numerical experiments (presented in the supplementary material) illustrate our theoretical findings. In particular, we provide examples of Lipschitz MDPs, for which the regret under DEL does not seem to scale with $S$ and $A$, and significantly outperforms algorithms that do not exploit the structure.

## 2 Related Work

Regret lower bounds have been extensively investigated for unstructured ergodic RL problems. [Burnetas and Katehakis, 1997] provided a problem-specific lower bound similar to ours, but only valid when the reward function is known. Minimax regret lower bounds have been studied e.g. in [Auer et al., 2009] and [Bartlett and Tewari, 2009]: in the worst case, the regret has to scale as $\sqrt{DSAT}$ where $D$ is the diameter of the MDP. In spite of these results, regret lower bounds for unstructured RL problems are still attracting some attention, see e.g. [Osband and Van Roy, 2016] for insightful discussions. To our knowledge, this paper constitutes the first attempt to derive regret lower bounds in the case of structured RL problems. Our bounds are asymptotic in the time horizon $T$, but we hope to extend them to finite time horizons using similar techniques as those recently used to provide such bounds for bandit problems [Garivier et al., Jun. 2018]. These techniques address problem-specific and minimax lower bounds in a unified manner, and can be leveraged to derive minimax lower bounds for structured RL problems. However we do not expect minimax lower bounds to be very informative about the regret gains that one may achieve by exploiting a structure (indeed, the MDPs leading to worst-case regret in unstructured RL comply to many structures).

There have been a plethora of algorithms developed for ergodic unstructured RL problems. We may classify these algorithms depending on their regret guarantees, either scaling as $\log T$ or $\sqrt{T}$. In absence of structure, [Burnetas and Katehakis, 1997] developed an asymptotically optimal, but involved, algorithm. This algorithm has been simplified in [Tewari and Bartlett, 2008], but remains

more complex than our proposed algorithm. Some algorithms have finite-time regret guarantees scaling as $\log T$ [Auer and Ortner, 2007], [Auer et al., 2009], [Filippi et al., 2010]. For example, the authors of [Filippi et al., 2010] propose KL-UCRL an extension of UCRL [Auer and Ortner, 2007] with regret bounded by $\frac{D^2 S^2 A}{\delta_{\min}} \log T$. Having finite-time regret guarantees is arguably desirable, but so far this comes at the expense of a much larger constant in front of $\log T$. Algorithms with regret scaling as $\sqrt{T}$ include UCRL2 [Auer et al., 2009], KL-UCRL with regret guarantees $\tilde{O}(DS\sqrt{AT})$, REGAL.C [Bartlett and Tewari, 2009] with guarantees $\tilde{O}(HS\sqrt{AT})$. Recently, the authors of [Agrawal and Jia, 2017] managed to achieve a regret guarantee of $\tilde{O}(D\sqrt{SAT})$, but only valid when $T \geq S^5 A$.

Algorithms devised to exploit some known structure are most often applicable to RL problems with continuous state or action spaces. Typically, the transition probabilities and reward function are assumed to be smooth in the state and action, typically Lipschitz continuous [Ortner and Ryabko, 2012], [Lakshmanan et al., 2015]. The regret then needs to scale as a power of $T$, e.g. $T^{2/3}$ in [Lakshmanan et al., 2015] for 1-dimensional state spaces. An original approach to RL problems for which the transition probabilities belong to some known class of functions was proposed in [Osband and Van Roy, 2014]. The regret upper bounds derived there depend on the so-called Kolmogorov and eluder dimensions, which in turn depend on the chosen class of functions. Our approach to design learning algorithms exploiting the structure is different from all aforementioned methods, as we aim at matching the problem-specific minimal exploration rates of sub-optimal (state, action) pairs.

## 3   Models and Objectives

We consider an MDP $\phi = (p_\phi, q_\phi)$ with finite state and action spaces $\mathcal{S}$ and $\mathcal{A}$ of respective cardinalities $S$ and $A$. $p_\phi$ and $q_\phi$ are the transition and reward kernels of $\phi$. Specifically, when in state $x$, taking action $a$, the system moves to state $y$ with probability $p_\phi(y|x,a)$, and a reward drawn from distribution $q_\phi(\cdot|x,a)$ of average $r_\phi(x,a)$ is collected. The rewards are bounded, w.l.o.g., in $[0,1]$. We assume that for any $(x,a)$, $q_\phi(\cdot|x,a)$ is absolutely continuous w.r.t. some measure $\lambda$ on $[0,1]$[2].

The random vector $Z_t := (X_t, A_t, R_t)$ represents the state, the action, and the collected reward at step $t$. A policy $\pi$ selects an action, denoted by $\pi_t(x)$, in step $t$ when the system is in state $x$ based on the history captured through $\mathcal{H}_t^\pi$, the $\sigma$-algebra generated by $(Z_1, \ldots, Z_{t-1}, X_t)$ observed under $\pi$: $\pi_t(x)$ is $\mathcal{H}_t^\pi$-measurable. We denote by $\Pi$ the set of all such policies.

**Structured MDPs.** The MDP $\phi$ is initially unknown. However we assume that $\phi$ belongs to some well specified set $\Phi$ which may encode a known structure of the MDP. The knowledge of $\Phi$ can be exploited to devise (more) efficient policies. The results derived in this paper are valid under *any* structure, but we give a particular attention to the cases of
*(i) Unstructured MDPs*: $\phi \in \Phi$ if for all $(x,a)$, $p_\phi(\cdot \mid x,a) \in \mathcal{P}(\mathcal{S})$ and $q_\phi(\cdot \mid x,a) \in \mathcal{P}([0,1])$[3];
*(ii) Lipschitz MDPs*: $\phi \in \Phi$ if $p_\phi(\cdot|x,a)$ and $r_\phi(x,a)$ are Lipschitz-continuous w.r.t. $x$ and $a$ in some metric space (we provide a precise definition in the next section).

**The learning problem.** The expected cumulative reward up to step $T$ of a policy $\pi \in \Pi$ when the system starts in state $x$ is $V_T^\pi(x) := \mathbb{E}_x^\pi[\sum_{t=1}^T R_t]$, where $\mathbb{E}_x^\pi[\cdot]$ denotes the expectation under policy $\pi$ given that $X_1 = x$. Now assume that the system starts in state $x$ and evolves according to the initially unknown MDP $\phi \in \Phi$ for given structure $\Phi$, the objective is to devise a policy $\pi \in \Pi$ maximizing $V_T^\pi(x)$ or equivalently, minimizing the regret $R_T^\pi(x)$ up to step $T$ defined as the difference between the cumulative reward of an optimal policy and that obtained under $\pi$:

$$R_T^\pi(x) := V_T^*(x) - V_T^\pi(x)$$

where $V_T^*(x) := \sup_{\pi \in \Pi} V_T^\pi(x)$.

**Preliminaries and notations.** Let $\Pi_D$ be the set of stationary (deterministic) policies, i.e. when in state $X_t = x$, $f \in \Pi_D$ selects an action $f(x)$ independent of $t$. $\phi$ is communicating if each pair of states are connected by some policy. Further, $\phi$ is ergodic if under any stationary policy,

the resulting Markov chain $(X_t)_{t \geq 1}$ is irreducible. For any communicating $\phi$ and any policy $\pi \in \Pi_D$, we denote by $g_\phi^\pi(x)$ the *gain* of $\pi$ (or long-term average reward) started from initial state $x$: $g_\phi^\pi(x) := \lim_{T \to \infty} \frac{1}{T} V_T^\pi(x)$. We denote by $\Pi^*(\phi)$ the set of stationary policies with maximal gain: $\Pi^*(\phi) := \{f \in \Pi_D : g_\phi^f(x) = g_\phi^*(x) \ \forall x \in \mathcal{S}\}$, where $g_\phi^*(x) := \max_{\pi \in \Pi} g_\phi^\pi(x)$. If $\phi$ is communicating, the maximal gain is constant and denoted by $g_\phi^*$. The *bias function* $h_\phi^f$ of $f \in \Pi_D$ is defined by $h_\phi^f(x) := C\text{-}\lim_{T \to \infty} \mathbb{E}_x^f[\sum_{t=1}^\infty (R_t - g_\phi^f(X_t))]$, and quantifies the advantage of starting in state $x$. We denote by $\mathbf{B}_\phi^a$ and $\mathbf{B}_\phi^*$, respectively, the Bellman operator under action $a$ and the optimal Bellman operator under $\phi$. They are defined by: for any $h : \mathcal{S} \mapsto \mathbb{R}$ and $x \in \mathcal{S}$,

$$(\mathbf{B}_\phi^a h)(x) := r_\phi(x, a) + \sum_{y \in \mathcal{S}} p_\phi(y|x, a) h(y) \quad \text{and} \quad (\mathbf{B}_\phi^* h)(x) := \max_{a \in \mathcal{A}} (\mathbf{B}_\phi^a h)(x) .$$

Then for any $f \in \Pi_D$, $g_\phi^f$ and $h_\phi^f$ satisfy the *evaluation equation*: for all state $x \in \mathcal{S}$, $g_\phi^f(x) + h_\phi^f(x) = (\mathbf{B}_\phi^{f(x)} h_\phi^f)(x)$. Furthermore, $f \in \Pi^*(\phi)$ if and only if $g_\phi^f$ and $h_\phi^f$ verify the *optimality equation*:

$$g_\phi^f(x) + h_\phi^f(x) = (\mathbf{B}_\phi^* h_\phi^f)(x) .$$

We denote by $h_\phi^*$ the bias function of an optimal stationary policy[4], and by $H$ its span $H := \max_{x,y} h_\phi^*(x) - h_\phi^*(y)$. For $x \in \mathcal{S}$, $h : \mathcal{S} \mapsto \mathbb{R}$, and $\phi \in \Phi$, let $\mathcal{O}(x; h, \phi) = \{a \in \mathcal{A} : (\mathbf{B}_\phi^* h)(x) = (\mathbf{B}_\phi^a h)(x)\}$. For ergodic $\phi$, $h_\phi^*$ is unique up to an additive constant. Hence, for ergodic $\phi$, the set of optimal actions in state $x$ under $\phi$ is $\mathcal{O}(x; \phi) := \mathcal{O}(x; h_\phi^*, \phi)$, and $\Pi^*(\phi) = \{f \in \Pi_D : f(x) \in \mathcal{O}(x; \phi) \ \forall x \in \mathcal{S}\}$. Finally, we define for any state $x$ and action $a$,

$$\delta^*(x, a; \phi) := (\mathbf{B}_\phi^* h_\phi^*)(x) - (\mathbf{B}_\phi^a h_\phi^*)(x) .$$

This can be interpreted as the long-term regret obtained by initially selecting action $a$ in state $x$ (and then applying an optimal stationary policy) rather than following an optimal policy. The minimum gap is defined as $\delta_{\min} := \min_{(x,a):\delta^*(x,a;\phi)>0} \delta^*(x, a; \phi)$.

We denote by $\bar{\mathbb{R}}_+ = \mathbb{R}_+ \cup \{\infty\}$. The set of MDPs is equipped with the following $\ell_\infty$-norm: $\|\phi - \psi\| := \max_{(x,a) \in \mathcal{S} \times \mathcal{A}} \|\phi(x, a) - \psi(x, a)\|$ where $\|\phi(x, a) - \psi(x, a)\| := |r_\phi(x, a) - r_\psi(x, a)| + \max_{y \in \mathcal{S}} |p_\phi(y \mid x, a) - p_\psi(y \mid x, a)|$.

The proofs of all results are presented in the supplementary material.

## 4   Regret Lower Bounds

In this section, we present an (asymptotic) regret lower bound satisfied by any *uniformly good* learning algorithm. An algorithm $\pi \in \Pi$ is uniformly good if for all ergodic $\phi \in \Phi$, any initial state $x$ and any constant $\alpha > 0$, the regret of $\pi$ satisfies $R_T^\pi(x) = o(T^\alpha)$ .

To state our lower bound, we introduce the following notations. For $\phi$ and $\psi$, we denote $\phi \ll \psi$ if the kernel of $\phi$ is absolutely continuous w.r.t. that of $\psi$, i.e., $\forall \mathcal{E}, \mathbb{P}_\phi[\mathcal{E}] = 0$ if $\mathbb{P}_\psi[\mathcal{E}] = 0$. For $\phi$ and $\psi$ such that $\phi \ll \psi$ and $(x, a)$, we define the KL-divergence between $\phi$ and $\psi$ in state-action pair $(x, a)$ $\text{KL}_{\phi|\psi}(x, a)$ as the KL-divergence between the distributions of the next state and collected reward if the state is $x$ and $a$ is selected under these two MDPs:

$$\text{KL}_{\phi|\psi}(x, a) = \sum_{y \in \mathcal{S}} p_\phi(y|x, a) \log \frac{p_\phi(y|x, a)}{p_\psi(y|x, a)} + \int_0^1 q_\phi(r|x, a) \log \frac{q_\phi(r|x, a)}{q_\psi(r|x, a)} \lambda(dr) .$$

We further define the set of *confusing* MDPs as:

$$\Delta_\Phi(\phi) = \{\psi \in \Phi : \phi \ll \psi, \ (i) \ \text{KL}_{\phi|\psi}(x, a) = 0 \ \forall x, \forall a \in \mathcal{O}(x; \phi); \ (ii) \ \Pi^*(\phi) \cap \Pi^*(\psi) = \emptyset\} .$$

This set consists of MDP $\psi$'s that $(i)$ coincide with $\phi$ for state-action pairs where the actions are optimal (the kernels of $\phi$ and $\psi$ cannot be statistically distinguished under an optimal policy); and such that $(ii)$ the optimal policies under $\psi$ are not optimal under $\phi$.

**Theorem 1.** *Let $\phi \in \Phi$ be ergodic. For any uniformly good algorithm $\pi \in \Pi$ and for any $x \in \mathcal{S}$,*

$$\liminf_{T \to \infty} \frac{R_T^\pi(x)}{\log T} \geq K_\Phi(\phi), \tag{1}$$

*where $K_\Phi(\phi)$ is the value of the following optimization problem:*

$$\inf_{\eta \in \mathcal{F}_\Phi(\phi)} \sum_{(x,a) \in \mathcal{S} \times \mathcal{A}} \eta(x,a) \delta^*(x,a;\phi), \tag{2}$$

*where $\mathcal{F}_\Phi(\phi) := \{\eta \in \bar{\mathbb{R}}_+^{S \times A} : \sum_{(x,a) \in \mathcal{S} \times \mathcal{A}} \eta(x,a) \mathrm{KL}_{\phi|\psi}(x,a) \geq 1, \forall \psi \in \Delta_\Phi(\phi)\}$.*

The above theorem can be interpreted as follows. When selecting a sub-optimal action $a$ in state $x$, one has to pay a regret of $\delta^*(x,a;\phi)$. Then the minimal number of times any sub-optimal action $a$ in state $x$ has to be explored scales as $\eta^*(x,a) \log T$ where $\eta^*(x,a)$ solves the optimization problem (2). It is worth mentioning that our lower bound is tight, as we present in Section 5 an algorithm achieving this fundamental limit of regret.

The regret lower bound stated in Theorem 1 extends the problem-specific regret lower bound derived in [Burnetas and Katehakis, 1997] for unstructured ergodic MDPs with known reward function. Our lower bound is valid for unknown reward function, but also applies to any structure $\Phi$. Note however that at this point, it is only implicitly defined through the solution of (2), which seems difficult to solve. The optimization problem can actually be simplified, as shown later in this section, by providing useful structural properties of the feasibility set $\mathcal{F}_\Phi(\phi)$ depending on the structure considered. The simplification will be instrumental to quantify the gain that can be achieved when optimally exploiting the structure, as well as to design efficient algorithms.

In the following, the optimization problem: $\inf_{\eta \in \mathcal{F}} \sum_{(x,a) \in \mathcal{S} \times \mathcal{A}} \eta(x,a) \delta^*(x,a;\phi)$ is referred to as $P(\phi, \mathcal{F})$; so that $P(\phi, \mathcal{F}_\Phi(\phi))$ corresponds to (2).

The proof of Theorem 1 combines a characterization of the regret as a function of the number of times $N_T(x,a)$ up to step $T$ (state, action) pair $(x,a)$ is visited, and of the $\delta^*(x,a;\phi)$'s, and change-of-measure arguments as those recently used to prove in a very direct manner regret lower bounds in bandit optimization problems [Kaufmann et al., 2016]. More precisely, for any uniformly good algorithm $\pi$, and for any confusing MDP $\psi \in \Delta_\Phi(\phi)$, we show that the exploration rates required to statistically distinguish $\psi$ from $\phi$ satisfy $\liminf_{T \to \infty} \frac{1}{\log T} \sum_{(x,a) \in \mathcal{S} \times \mathcal{A}} \mathbb{E}_{x_1}^\pi[N_T(x,a)] \mathrm{KL}_{\phi|\psi}(x,a) \geq 1$ where the expectation is taken w.r.t. $\phi$ given any initial state $x_1$. The theorem is then obtained by considering (hence optimizing the lower bound) all possible confusing MDPs.

## 4.1 Decoupled exploration in unstructured MDPs

In the absence of structure, $\Phi = \{\psi : p_\psi(\cdot|x,a) \in \mathcal{P}(\mathcal{S}), q_\psi(\cdot|x,a) \in \mathcal{P}([0,1]), \forall(x,a)\}$, and we have:

**Theorem 2.** *Consider the unstructured model $\Phi$, and let $\phi \in \Phi$ be ergodic. We have:*

$$\mathcal{F}_\Phi(\phi) = \left\{\eta \in \bar{\mathbb{R}}_+^{S \times A} : \forall(x,a) \text{ s.t. } a \notin \mathcal{O}(x;\phi), \eta(x,a)\mathrm{KL}_{\phi|\psi}(x,a) \geq 1, \forall \psi \in \Delta_\Phi(x,a;\phi)\right\}$$

*where $\Delta_\Phi(x,a;\phi) := \{\psi \in \Phi : \mathrm{KL}_{\phi|\psi}(y,b) = 0 \,\forall(y,b) \neq (x,a) \text{ and } (\mathbf{B}_\psi^a h_\phi^*)(x) > g_\phi^* + h_\phi^*(x)\}$.*

The theorem states that in the constraints of the optimization problem (2), we can restrict our attention to confusing MDPs $\psi$ that are different than the original MDP $\phi$ only for a single state-action pair $(x,a)$. Further note that the condition $(\mathbf{B}_\psi^a h_\phi^*)(x) > g_\phi^* + h_\phi^*(x)$ is equivalent to saying that action $a$ becomes optimal in state $x$ under $\psi$ (see Lemma 1(i) in [Burnetas and Katehakis, 1997]). Hence to obtain the lower bound in unstructured MDPs, we may just consider confusing MDPs $\psi$ which make an initially sub-optimal action $a$ in state $x$ optimal by locally changing the kernels and rewards of $\phi$ at $(x,a)$ only. Importantly, this observation implies that an optimal algorithm $\pi$ must satisfy $\mathbb{E}_{x_1}^\pi[N_T(x,a)] \sim \log T / \inf_{\psi \in \Delta_\Phi(x,a;\phi)} \mathrm{KL}_{\phi|\psi}(x,a)$. In other words, the required level of exploration of the various sub-optimal state-action pairs are *decoupled*, which significantly simplifies the design of optimal algorithms.

To get an idea on how the regret lower bound scales as the sizes of both state and action spaces, we can further provide an upper bound of the regret lower bound. One may easily observe that

$\mathcal{F}_{\mathrm{un}}(\phi) \subset \mathcal{F}_{\Phi}(\phi)$ where

$$\mathcal{F}_{\mathrm{un}}(\phi) = \left\{ \eta \in \bar{\mathbb{R}}_+^{S \times A} : \eta(x,a) \left( \frac{\delta^*(x,a;\phi)}{H+1} \right)^2 \geq 2, \; \forall (x,a) \text{ s.t. } a \notin \mathcal{O}(x;\phi) \right\} .$$

From this result, an upper bound of the regret lower bound is $K_{\mathrm{un}}(\phi) := 2\frac{(H+1)^2}{\delta_{\min}} SA \log T$, and we can devise algorithms achieving this regret scaling (see Section 5).

Theorem 2 relies on the following decoupling lemma, actually valid under any structure $\Phi$.

**Lemma 1.** *Let $\mathcal{U}_1, \mathcal{U}_2$ be two non-overlapping subsets of the (state, action) pairs such that for all $(x,a) \in \mathcal{U}_0 := \mathcal{U}_1 \cup \mathcal{U}_2$, $a \notin \mathcal{O}(x;\phi)$. Define the following three MDPs in $\Phi$ obtained starting from $\phi$ and changing the kernels for (state, action) pairs in $\mathcal{U}_1 \cup \mathcal{U}_2$. Specifically, let $(p,q)$ be some transition and reward kernels. For all $(x,a)$, define $\psi_j$, $j \in \{0,1,2\}$ as*

$$(p_{\psi_j}(\cdot|x,a), q_{\psi_j}(\cdot|x,a)) = \begin{cases} (p(\cdot|x,a), q(\cdot|x,a)) & \text{if } (x,a) \in \mathcal{U}_j, \\ (p_\phi(\cdot|x,a), q_\phi(\cdot|x,a)) & \text{otherwise.} \end{cases}$$

*Then, if $\Pi^*(\phi) \cap \Pi^*(\psi_0) = \emptyset$, then $\Pi^*(\phi) \cap \Pi^*(\psi_1) = \emptyset$ or $\Pi^*(\phi) \cap \Pi^*(\psi_2) = \emptyset$.*

### 4.2 Lipschitz structure

Lipschitz structures have been widely studied in the bandit and reinforcement learning literature. We find it convenient to use the following structure, although one could imagine other variants in more general metric spaces. We assume that the state (resp. action) space can be embedded in the $d$ (resp. $d'$) dimensional Euclidian space: $\mathcal{S} \subset [0,D]^d$ and $\mathcal{A} \subset [0,D']^{d'}$. We consider MDPs whose transition kernels and average rewards are Lipschitz w.r.t. the states and actions. Specifically, let $L, L' > 0$, $\alpha, \alpha' > 0$, and

$$\Phi = \{\psi : p_\psi(\cdot|x,a) \in \mathcal{P}(\mathcal{S}), q_\psi(\cdot|x,a) \in \mathcal{P}([0,1]) : (L1)\text{-}(L2) \text{ hold}, \forall (x,a)\},$$

where

$$(L1) \qquad \|p_\psi(\cdot|x,a) - p_\psi(\cdot|x',a')\|_1 \leq Ld(x,x')^\alpha + L'd(a,a')^{\alpha'} ,$$

$$(L2) \qquad |r_\psi(x,a) - r_\psi(x',a')| \leq Ld(x,x')^\alpha + L'd(a,a')^{\alpha'} .$$

Here $d(\cdot,\cdot)$ is the Euclidean distance, and for two distributions $p_1$ and $p_2$ on $\mathcal{S}$ we denote by $\|p_1 - p_2\|_1 = \sum_{y \in \mathcal{S}} |p_1(y) - p_2(y)|$.

**Theorem 3.** *For the model $\Phi$ with Lipschitz structure (L1)-(L2), we have $\mathcal{F}_{\mathrm{lip}}(\phi) \subset \mathcal{F}_{\Phi}(\phi)$ where $\mathcal{F}_{\mathrm{lip}}(\phi)$ is the set of $\eta \in \bar{\mathbb{R}}_+^{S \times A}$ satisfying for all $(x',a')$ such that $a' \notin \mathcal{O}(x',\phi)$,*

$$\sum_{x \in \mathcal{S}} \sum_{a \notin \mathcal{O}(x,\phi)} \eta(x,a) \left( \left[ \frac{\delta^*(x',a';\phi)}{H+1} - 2\left(Ld(x,x')^\alpha + L'd(a,a')^{\alpha'}\right) \right]_+ \right)^2 \geq 2 \qquad (3)$$

*where we use the notation $[u]_+ := \max\{0,u\}$ for $u \in \mathbb{R}$. Furthermore, the optimal values $K_\Phi(\phi)$ and $K_{\mathrm{lip}}(\phi)$ of $P(\phi, \mathcal{F}_\Phi(\phi))$ and $P(\phi, \mathcal{F}_{\mathrm{lip}}(\phi))$ are upper bounded by $8\frac{(H+1)^3}{\delta_{\min}^2} S_{\mathrm{lip}} A_{\mathrm{lip}}$ where*

$$S_{\mathrm{lip}} := \min \left\{ S, \left( \frac{D\sqrt{d}}{\left(\frac{\delta_{\min}}{8L(H+1)}\right)^{1/\alpha}} + 1 \right)^d \right\}, \text{ and } A_{\mathrm{lip}} := \min \left\{ A, \left( \frac{D'\sqrt{d'}}{\left(\frac{\delta_{\min}}{8L'(H+1)}\right)^{1/\alpha'}} + 1 \right)^{d'} \right\}.$$

The above theorem has important consequences. First, it states that exploiting the Lipschitz structure optimally, one may achieve a regret at most scaling as $\frac{(H+1)^3}{\delta_{\min}^2} S_{\mathrm{lip}} A_{\mathrm{lip}} \log T$. This scaling is independent of the sizes of the state and action spaces provided that the minimal gap $\delta_{\min}$ is fixed, and provided that the span $H$ does not scale with $S$. The latter condition typically holds for fast mixing models or for MDPs with diameter not scaling with $S$ (refer to [Bartlett and Tewari, 2009] for a precise connection between $H$ and the diameter). Hence, exploiting the structure can really yield significant regret improvements. As shown in the next section, leveraging the simplified structure in $\mathcal{F}_{\mathrm{lip}}(\phi)$, we may devise a simple algorithm achieving these improvements, i.e., having a regret scaling at most as $K_{\mathrm{lip}}(\phi) \log T$.

---
**Algorithm 1** DEL($\gamma$)
---
**input** Model structure $\Phi$

Initialize $N_1(x) \leftarrow \mathbb{1}[x = X_1]$, $N_1(x, a) \leftarrow 0$, $s_1(x) \leftarrow 0$, $p_1(y \mid x, a) \leftarrow 1/|\mathcal{S}|$, $r_1(x, a) \leftarrow 0$
for each $x, y \in \mathcal{S}$, $a \in \mathcal{A}$, and $\phi_1$ accordingly.

**for** $t = 1, ..., T$ **do**

    For each $x \in \mathcal{S}$, let $\mathcal{C}_t(x) := \{a \in \mathcal{A} : N_t(x, a) \geq \log^2 N_t(x)\}$, $\phi'_t := \phi_t(\mathcal{C}_t)$, $h'_t(x) := h^*_{\phi'_t}(x)$, $\zeta_t := \frac{1}{1 + \log \log t}$ and $\gamma_t := (1 + \gamma)(1 + \log t)$

    **if** $\forall a \in \mathcal{O}(x; \phi'_t)$, $N_t(X_t, a) < \log^2 N_t(X_t) + 1$ **then**

        **Monotonize:** $A_t \leftarrow A_t^{\mathrm{mnt}} := \arg\min_{a \in \mathcal{O}(x; \phi'_t)} N_t(X_t, a)$.

    **else if** $\exists a \in \mathcal{A}$ s.t. $N_t(X_t, a) < \frac{\log N_t(X_t)}{1 + \log \log N_t(X_t)}$ **then**

        **Estimate:** $A_t \leftarrow A_t^{\mathrm{est}} := \arg\min_{a \in \mathcal{A}} N_t(X_t, a)$.

    **else if** $\left( \frac{N_t(x,a)}{\gamma_t} : (x, a) \in \mathcal{S} \times \mathcal{A} \right) \in \mathcal{F}_\Phi(\phi_t; \mathcal{C}_t, \zeta_t)$. **then**

        **Exploit:** $A_t \leftarrow A_t^{\mathrm{xpt}} := \arg\min_{a \in \mathcal{O}(x; \phi'_t)} N_t(X_t, a)$.

    **else**

        For each $(x, a) \in \mathcal{S} \times \mathcal{A}$, let $\delta_t(x, a) := \delta^*(x, a; \phi_t, \mathcal{C}_t, \zeta_t)$.

        **if** $\mathcal{F}_t := \mathcal{F}_\Phi(\phi_t; \mathcal{C}_t, \zeta_t) \cap \{\eta : \eta(x, a) = \infty \text{ if } \delta_t(x, a) = 0\} = \emptyset$ **then**

            Let $\eta_t(x, a) = \infty$ if $\delta_t(x, a) = 0$ and $\eta_t(x, a) = 0$ otherwise.

        **else**

            Obtain a solution $\eta_t$ of $\mathcal{P}(\delta_t, \mathcal{F}_t)$: $\inf_{\eta \in \mathcal{F}_t} \sum_{(x,a) \in \mathcal{S} \times \mathcal{A}} \eta(x, a) \delta_t(x, a)$

        **end if**

        **Explore:** $A_t \leftarrow A_t^{\mathrm{xpr}} := \arg\min_{a \in \mathcal{A} : N_t(X_t, a) \leq \eta_t(X_t, a) \gamma_t} N_t(X_t, a)$.

    **end if**

    Select action $A_t$, and observe the next state $X_{t+1}$ and the instantaneous reward $R_t$.

    Update $\phi_{t+1}$, $N_{t+1}(x)$ and $N_{t+1}(x, a)$ for each $(x, a) \in \mathcal{S} \times \mathcal{A}$.

**end for**

---

## 5 Algorithms

In this section, we present DEL (Directed Exploration Learning), an algorithm that achieves the regret limits identified in the previous section. Asymptotically optimal algorithms for generic controlled Markov chains have already been proposed in [Graves and Lai, 1997], and could be adapted to our setting. By presenting DEL, we aim at providing simplified, yet optimal algorithms. Moreover, DEL can be adapted so that the exploration rates of sub-optimal actions are *directed* towards the solution of an optimization problem $P(\phi, \mathcal{F}(\phi))$ provided that $\mathcal{F}(\phi) \subset \mathcal{F}_\Phi(\phi)$ (it suffices to use $\mathcal{F}(\phi_t)$ instead of $\mathcal{F}_\Phi(\phi_t)$ in DEL). For example, in the case of Lipschitz structure $\Phi$, running DEL on $\mathcal{F}_{\mathrm{lip}}(\phi)$ yields a regret scaling at most as $\frac{(H+1)^3}{\delta_{\min}^2} S_{\mathrm{lip}} A_{\mathrm{lip}} \log T$.

The pseudo-code of DEL with input parameter $\gamma > 0$ is given in Algorithm 1. There, for notational convenience, we abuse the notations and redefine $\log t$ as $\mathbb{1}[t \geq 1] \log t$, and let $\infty \cdot 0 = 0$. $\phi_t$ refers to the estimated MDP at time $t$ (using empirical transition rates and rewards). For any non-empty correspondence $\mathcal{C} : \mathcal{S} \twoheadrightarrow \mathcal{A}$ (i.e., for any $x$, $\mathcal{C}(x)$ is a non-empty subset of $\mathcal{A}$), let $\phi(\mathcal{C})$ denote the restricted MDP where the set of actions available at state $x$ is $\mathcal{C}(x)$. Then, $g^*_{\phi(\mathcal{C})}$ and $h^*_{\phi(\mathcal{C})}$ are the (optimal) gain and bias functions corresponding to the restricted MDP $\phi(\mathcal{C})$. Given a restriction defined by $\mathcal{C}$, for each $(x, a) \in \mathcal{S} \times \mathcal{A}$, let $\delta^*(x, a; \phi, \mathcal{C}) := (\mathbf{B}^*_{\phi(\mathcal{C})} h^*_{\phi(\mathcal{C})})(x) - (\mathbf{B}^a_\phi h^*_{\phi(\mathcal{C})})(x)$ and $H_{\phi(\mathcal{C})} := \max_{x,y \in \mathcal{S}} h^*_{\phi(\mathcal{C})}(x) - h^*_{\phi(\mathcal{C})}(y)$. For $\zeta \geq 0$, let $\delta^*(x, a; \phi, \mathcal{C}, \zeta) := 0$ if $\delta^*(x, a; \phi, \mathcal{C}) \leq \zeta$, and let $\delta^*(x, a; \phi, \mathcal{C}, \zeta) := \delta^*(x, a; \phi, \mathcal{C})$ otherwise. For $\zeta \geq 0$, we further define the set of confusing MDPs $\Delta_\Phi(\phi; \mathcal{C}, \zeta)$, and the set of feasible solutions $\mathcal{F}_\Phi(\phi; \mathcal{C}, \zeta)$ as:

$$\Delta_\Phi(\phi; \mathcal{C}, \zeta) := \left\{ \psi \in \Phi \cup \{\phi\} : \phi \ll \psi; \begin{array}{l} (i)\ \mathrm{KL}_{\phi|\psi}(x, a) = 0\ \forall x, \forall a \in \mathcal{O}(x; \phi(\mathcal{C})); \\ (ii)\ \exists (x, a) \in \mathcal{S} \times \mathcal{A}\ \text{s.t.} \\ \qquad a \notin \mathcal{O}(x; \phi(\mathcal{C}))\ \text{and}\ \delta^*(x, a; \psi, \mathcal{C}, \zeta) = 0 \end{array} \right\}$$

$$\mathcal{F}_\Phi(\phi; \mathcal{C}, \zeta) := \left\{ \eta \in \bar{\mathbb{R}}_+^{S \times A} : \sum_{x \in \mathcal{S}} \sum_{a \in \mathcal{A}} \eta(x, a) \mathrm{KL}_{\phi|\psi}(x, a) \geq 1,\ \forall \psi \in \Delta_\Phi(\phi; \mathcal{C}, \zeta) \right\}.$$

Similar sets $\mathcal{F}_{\text{un}}(\phi; \mathcal{C}, \zeta)$ and $\mathcal{F}_{\text{lip}}(\phi; \mathcal{C}, \zeta)$ can be defined for the cases of unstructured and Lipschitz MDPs (refer to the supplementary material), and DEL can be simplified in these cases by replacing $\mathcal{F}_{\Phi}(\phi; \mathcal{C}, \zeta)$ by $\mathcal{F}_{\text{un}}(\phi; \mathcal{C}, \zeta)$ or $\mathcal{F}_{\text{lip}}(\phi; \mathcal{C}, \zeta)$ in the pseudo-code. Finally, $\mathcal{P}(\delta, \mathcal{F})$ refers to the optimization problem $\inf_{\eta \in \mathcal{F}} \sum_{(x,a) \in \mathcal{S} \times \mathcal{A}} \eta(x, a) \delta(x, a)$.

DEL combines the ideas behind OSSB [Combes et al., 2017], an asymptotically optimal algorithm for structured bandits, and the asymptotically optimal algorithm presented in [Burnetas and Katehakis, 1997] for RL problems without structure. DEL design aims at exploring sub-optimal actions no more than what the regret lower bound prescribes. To this aim, it essentially solves in each iteration $t$ an optimization problem close to $P(\phi_t, \mathcal{F}_{\Phi}(\phi_t))$ where $\phi_t$ is an estimate of the true MDP $\phi$. Depending on the solution and the number of times apparently sub-optimal actions have been played, DEL decides to explore or exploit. The estimation phase ensures that certainty equivalence holds. The "monotonization" phase together with the restriction to relatively well selected actions were already proposed in [Burnetas and Katehakis, 1997] to make sure that accurately estimated actions only are selected in the exploitation phase. The various details and complications introduced in DEL ensure that its regret analysis can be conducted. In practice (see the supplementary material), our initial experiments suggest that many details can be removed without large regret penalties.

**Theorem 4.** *For a structure $\Phi$ with Bernoulli rewards and for any ergodic MDP $\phi \in \Phi$, assume that: (i) $\phi$ is in the interior of $\Phi$ (i.e., there exists a constant $\zeta_0 > 0$ such that for any $\zeta \in (0, \zeta_0)$, $\psi \in \Phi$ if $\|\phi - \psi\| \leq \zeta$ and $\psi \ll \phi$), (ii) the solution $\eta^*(\phi)$ is uniquely defined for each $(x, a)$ such that $a \notin \mathcal{O}(x; \phi)$, (iii) continuous at $\phi$ (i.e., for any given $\varepsilon > 0$, there exists $\zeta(\varepsilon) > 0$ such that for all $\zeta \in (0, \zeta(\varepsilon))$, if $\|\psi - \phi\| \leq \zeta$, $\max_{(x,a):a \notin \mathcal{O}(x;\phi)} |\eta^*(x, a; \psi, \zeta) - \eta^*(x, a; \phi)| \leq \varepsilon$ where $\eta^*(\psi, \zeta)$ is solution of $\mathcal{P}(\delta^*(\psi, \mathcal{A}, \zeta), \mathcal{F}_{\Phi}(\psi; \mathcal{A}, \zeta))$, and $\eta^*(x, a; \phi)$ that of $P(\phi, \mathcal{F}_{\Phi}(\phi)))$. Then, for $\pi = \mathrm{DEL}(\gamma)$ with any $\gamma > 0$, we have:*

$$\limsup_{T \to \infty} \frac{R_T^\pi(\phi)}{\log T} \leq (1 + \gamma) K_\Phi(\phi) . \tag{4}$$

*For Lipschitz $\Phi$ with (L1)-(L2) (resp. unstructured $\Phi$), if $\pi = \mathrm{DEL}$ uses in each step $t$, $\mathcal{F}_{\text{lip}}(\phi_t; \mathcal{C}_t, \zeta_t)$ (resp. $\mathcal{F}_{\text{un}}(\phi_t; \mathcal{C}_t, \zeta_t)$) instead of $\mathcal{F}_{\Phi}(\phi_t; \mathcal{C}_t, \zeta_t)$, its regret is asymptotically smaller than $(1 + \gamma) K_{\text{lip}}(\phi) \log T$ (resp. $(1 + \gamma) K_{\text{un}}(\phi) \log T$).*

In the above theorem, the assumptions about the uniqueness and continuity of the solution $\eta^*(\phi)$ could be verified for particular structures. In particular, we believe that they generally hold in the case of unstructured and Lipschitz MDPs. Also note that similar assumptions have been made in [Graves and Lai, 1997].

## 6 Extensions and Future Work

It is worth extending the approach developed in this paper to the case of structured discounted RL problems (although for such problems, there is no ideal way of defining the regret of an algorithm). There are other extensions worth investigating. For example, since our framework allows any kind of structure, we may specify our regret lower bounds for structures stronger than that corresponding to Lipschitz continuity, e.g., the reward may exhibit some kind of unimodality or convexity. Under such structures, the regret improvements might become even more significant. Another interesting direction consists in generalizing the results to the case of communicating MDPs. This would allow us for example to consider deterministic system dynamics and unknown probabilistic rewards.

## 7 Acknowledgements

This work was partially supported by the Wallenberg AI, Autonomous Systems and Software Program (WASP) funded by the Knut and Alice Wallenberg Foundation. Jungseul Ok is now with UIUC in Prof. Sewoong Oh's group. He would like to thank UIUC for financially supporting his participation to NIPS 2018 conference.

## Footnotes

[1]The first lower bound is asymptotic in $T$ and problem-specific, the second is minimax. We ignore here for simplicity the dependence of these bounds in the diameter, bias span, and action sub-optimality gap.

[2]$\lambda$ can be the Lebesgue measure; alternatively, if rewards take values in $\{0,1\}$, $\lambda$ can be the sum of Dirac measures at 0 and 1.

[3]$\mathcal{P}(\mathcal{S})$ is the set of distributions on $\mathcal{S}$ and $\mathcal{P}([0,1])$ is the set of distributions on $[0,1]$, absolutely continuous w.r.t. $\lambda$.

[4]In case of $h_\phi^*$ is not unique, we arbitrarily select an optimal stationary policy and define $h_\phi^*$.

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
