[Supplementary Material]

# Exploration in Structured Reinforcement Learning (Supplementary Material)

**Jungseul Ok**
KTH, EECS
Stockholm, Sweden
ockjs@illinois.edu

**Alexandre Proutiere**
KTH, EECS
Stockholm, Sweden
alepro@kth.se

**Damianos Tranos**
KTH, EECS
Stockholm, Sweden
tranos@kth.se

## A   The DEL Algorithm

In this section, we present DEL, our algorithm, and introduce notations used to describe it and in its regret analysis. DEL pseudo-code is given in Algorithm S1. There, for notational convenience, we abuse the notations and redefine $\log t$ as $\mathbb{1}[t \geq 1] \log t$. $\phi_t$ refers to the estimated MDP at time $t$ (e.g. using empirical transition rates). For non-empty correspondence $\mathcal{C} : \mathcal{S} \twoheadrightarrow \mathcal{A}$ (i.e., for any $x$, $\mathcal{C}(x)$ is a non-empty subset of $\mathcal{A}$), let $\phi(\mathcal{C})$ denote the restricted MDP where the set of actions available at state $x$ is limited to $\mathcal{C}(x)$. Then, $g^*_{\phi(\mathcal{C})}$ and $h^*_{\phi(\mathcal{C})}$ are the (optimal) gain and bias functions corresponding to the restricted MDP $\phi(\mathcal{C})$, respectively. Given a restriction defined by $\mathcal{C}$, for each $(x, a) \in \mathcal{S} \times \mathcal{A}$, let $\delta^*(x, a; \phi, \mathcal{C}) := (\mathbf{B}^*_{\phi(\mathcal{C})} h^*_{\phi(\mathcal{C})})(x) - (\mathbf{B}^a_\phi h^*_{\phi(\mathcal{C})})(x)$ and $H_{\phi(\mathcal{C})} := \max_{x,y \in \mathcal{S}} h^*_{\phi(\mathcal{C})}(x) - h^*_{\phi(\mathcal{C})}(y)$. For $\zeta \geq 0$, let $\delta^*(x, a; \phi, \mathcal{C}, \zeta) := 0$ if $\delta^*(x, a; \phi, \mathcal{C}) \leq \zeta$, and let $\delta^*(x, a; \phi, \mathcal{C}, \zeta) := \delta^*(x, a; \phi, \mathcal{C})$ otherwise. For $\zeta \geq 0$, we further define the set of confusing MDPs $\Delta_\Phi(\phi; \mathcal{C}, \zeta)$, and the set of feasible solutions $\mathcal{F}_\Phi(\phi; \mathcal{C}, \zeta)$:

$$\Delta_\Phi(\phi; \mathcal{C}, \zeta) := \left\{ \psi \in \Phi \cup \{\phi\} : \phi \ll \psi; \begin{array}{l} (i)\ \text{KL}_{\phi|\psi}(x, a) = 0 \ \forall x, \forall a \in \mathcal{O}(x; \phi(\mathcal{C})); \\ (ii)\ \exists (x, a) \in \mathcal{S} \times \mathcal{A} \text{ s.t.} \\ \quad a \notin \mathcal{O}(x; \phi(\mathcal{C})) \text{ and } \delta^*(x, a; \psi, \mathcal{C}, \zeta) = 0 \end{array} \right\}$$

$$\mathcal{F}_\Phi(\phi; \mathcal{C}, \zeta) := \left\{ \eta \in \bar{\mathbb{R}}^{S \times A}_+ : \sum_{x \in \mathcal{S}} \sum_{a \in \mathcal{A}} \eta(x, a) \text{KL}_{\phi|\psi}(x, a) \geq 1 \ \forall \psi \in \Delta_\Phi(\phi; \mathcal{C}, \zeta) \right\}.$$

For the unstructured and Lipschitz MDPs, we simplify the feasible solution set as $\mathcal{F}_{\text{un}}(\phi; \mathcal{C}, \zeta)$ and $\mathcal{F}_{\text{lip}}(\phi; \mathcal{C}, \zeta)$, respectively, defined as:

$$\mathcal{F}_{\text{un}}(\phi; \mathcal{C}, \zeta) := \left\{ \eta \in \bar{\mathbb{R}}^{S \times A}_+ : \eta(x, a) \left( \frac{\delta^*(x, a; \phi, \mathcal{C}, \zeta)}{H_{\phi(\mathcal{C})} + 1} \right)^2 \geq 2 \ \forall (x, a) \text{ s.t. } a \notin \mathcal{O}(x; \phi(\mathcal{C})) \right\}$$

$$\mathcal{F}_{\text{lip}}(\phi; \mathcal{C}, \zeta) := \left\{ \eta \in \bar{\mathbb{R}}^{S \times A}_+ : L_{\text{lip}}(x', a'; \phi, \mathcal{C}, \zeta) \geq 2 \forall (x', a') \text{ s.t. } a' \notin \mathcal{O}(x'; \phi(\mathcal{C})) \right\}$$

where

$$L_{\text{lip}}(x', a'; \phi, \mathcal{C}, \zeta) := \sum_{x \in \mathcal{S}} \sum_{a \in \mathcal{A}} \eta(x, a) \left( \left[ \frac{\delta^*(x', a'; \phi, \mathcal{C}, \zeta)}{H_{\phi(\mathcal{C})} + 1} - 2 \left( L d(x, x')^\alpha + L' d(a, a')^{\alpha'} \right) \right]_+ \right)^2.$$

## B   Numerical Experiments

In this section, we briefly illustrate the performance of a simplified version of the DEL algorithm on a simple example constructed so as to comply to a Lipschitz structure. Our objective is to investigate

**Algorithm S1** DEL($\gamma$)

---

**input** Model structure $\Phi$

Initialize $N_1(x) \leftarrow \mathbb{1}[x = X_1]$, $N_1(x,a) \leftarrow 0$, $s_1(x) \leftarrow 0$, $p_1(y \mid x,a) \leftarrow 1/|\mathcal{S}|$, $r_1(x,a) \leftarrow 0$
for each $x, y \in \mathcal{S}$, $a \in \mathcal{A}$, and $\phi_1$ accordingly.

**for** $t = 1, ..., T$ **do**

For each $x \in \mathcal{S}$, let $\mathcal{C}_t(x) := \{a \in \mathcal{A} : N_t(x,a) \geq \log^2 N_t(x)\}$, $\phi_t' := \phi_t(\mathcal{C}_t)$, $h_t'(x) := h_{\phi_t'}^*(x)$, $\zeta_t := \frac{1}{1+\log\log t}$ and $\gamma_t := (1+\gamma)(1+\log t)$

**if** $\forall a \in \mathcal{O}(x; \phi_t')$, $N_t(X_t, a) < \log^2 N_t(X_t) + 1$ **then**

**Monotonize:** $A_t \leftarrow A_t^{\text{mnt}} := \arg\min_{a \in \mathcal{O}(x;\phi_t')} N_t(X_t, a)$.

**else if** $\exists a \in \mathcal{A}$ s.t. $N_t(X_t, a) < \frac{\log N_t(X_t)}{1+\log\log N_t(X_t)}$ **then**

**Estimate:** $A_t \leftarrow A_t^{\text{est}} := \arg\min_{a \in \mathcal{A}} N_t(X_t, a)$.

**else if** $\left( \frac{N_t(x,a)}{\gamma_t} : (x,a) \in \mathcal{S} \times \mathcal{A} \right) \in \mathcal{F}_\Phi(\phi_t; \mathcal{C}_t, \zeta_t)$. **then**

**Exploit:** $A_t \leftarrow A_t^{\text{xpt}} := \arg\min_{a \in \mathcal{O}(x;\phi_t')} N_t(X_t, a)$.

**else**

For each $(x,a) \in \mathcal{S} \times \mathcal{A}$, let $\delta_t(x,a) := \delta^*(x,a; \phi_t, \mathcal{C}_t, \zeta_t)$.

**if** $\mathcal{F}_t := \mathcal{F}_\Phi(\phi_t; \mathcal{C}_t, \zeta_t) \cap \{\eta : \eta(x,a) = \infty \text{ if } \delta_t(x,a) = 0\} = \emptyset$ **then**

Let $\eta_t(x,a) = \infty$ if $\delta_t(x,a) = 0$ and $\eta_t(x,a) = 0$ otherwise.

**else**

Obtain a solution $\eta_t$ of $\mathcal{P}(\delta_t, \mathcal{F}_t)$: $\inf_{\eta \in \mathcal{F}_t} \sum_{(x,a) \in \mathcal{S} \times \mathcal{A}} \eta(x,a)\delta_t(x,a)$

**end if**

**Explore:** $A_t \leftarrow A_t^{\text{xpr}} := \arg\min_{a \in \mathcal{A}: N_t(X_t,a) \leq \eta_t(X_t,a)\gamma_t} N_t(X_t, a)$.

**end if**

Select action $A_t$, and observe the next state $X_{t+1}$ and the instantaneous reward $R_t$.
Update $\phi_{t+1}$:

$$N_{t+1}(x) \leftarrow N_t(x) + \mathbb{1}[x = X_{t+1}],$$
$$N_{t+1}(x,a) \leftarrow N_t(x,a) + \mathbb{1}[(x,a) = (X_t, A_t)],$$
$$p_{t+1}(y \mid x,a) \leftarrow \begin{cases} \frac{N_t(x,a)p_t(y|x,a)+\mathbb{1}[y=X_{t+1}]}{N_{t+1}(x,a)} & \text{if } (x,a) = (X_t, A_t) \\ p_t(y \mid x,a) & \text{otherwise} \end{cases}, \quad \forall x, y \in \mathcal{S}, a \in \mathcal{A}$$
$$r_{t+1}(x,a) \leftarrow \begin{cases} \frac{N_t(x,a)r_t(x,a)+R_t}{N_{t+1}(x,a)} & \text{if } (x,a) = (X_t, A_t) \\ r_t(x,a) & \text{otherwise} \end{cases}$$

**end for**

---

the regret gains obtained by exploiting a Lipschitz structure, and we compare the performance of our two simplified versions of DEL with $\gamma = 1$ and $\zeta_t = 0$, one solving $P(\phi_t, \mathcal{F}_{\text{un}}(\phi_t; \mathcal{C}_t, \zeta_t))$ in step $t$, and the other solving $P(\phi_t, \mathcal{F}_{\text{lip}}(\phi_t; \mathcal{C}_t, \zeta_t))$.

**The RL problem.** We consider a toy MDP whose states are partitioned into two *clusters* $\mathcal{S}_1$, $\mathcal{S}_2$ of equal sizes $S/2$. Both states and actions are embedded into $\mathbb{R}$:

- The states in cluster $\mathcal{S}_1$ (resp. $\mathcal{S}_2$) are randomly generated in the interval $[-\zeta, 0]$ (resp. $[1, 1+\zeta]$) for some $\zeta \in (0,1)$;

- In each state there are two possible actions: $s = 0$ (stands for *stay*) and $m = 1$ (stands for *move*).

The transition probabilities depend on the states only through their corresponding clusters, and are given by: for $\epsilon \in (0, 0.5)$,

$$p(y|x,a) = \begin{cases} \frac{2(1-\epsilon)}{S} & \text{if } (x,y,a) \in \Gamma_p \\ \frac{2\epsilon}{S} & \text{otherwise} \end{cases}, \tag{S1}$$

where

$$\Gamma_p := \{(x,y,a) : a = s, \exists i \in \{1,2\}, x, y \in \mathcal{S}_i\} \cup \{(x,y,a) : a = m, \exists i \in \{1,2\}, x \in \mathcal{S}_i, y \notin \mathcal{S}_i\}.$$

In words, when the agent decides to move, she will end up in a state uniformly sampled from the other cluster with probability $1 - \epsilon$; when she decides to stay, she changes state within her cluster

uniformly at random. We take $\epsilon > 0$ to ensure irreducibility. For numerical experiments we take $\epsilon = 0.1$ and $\zeta = 0.1$. The reward is obtained according to the following deterministic rule:

$$r(x,a) = \begin{cases} 1 & \text{if } (x,a) : a = m \text{ and } x \in \mathcal{S}_1, \\ 0 & \text{otherwise.} \end{cases} \tag{S2}$$

A reward is collected when the agent is in cluster $\mathcal{S}_1$ and decides to move. The optimal stationary strategy consists in moving in each state.

(a) Regret over time

(b) Regret at $T = 50k$ varying $S$

Figure S1: Averaged regret under the two simplified versions of DEL over $48$ random samples: `Unstructured` (or `Un`) and `Lipchitz` (or `Lip`) refer to the algorithm with $\mathcal{F}_{\text{un}}$ and $\mathcal{F}_{\text{lip}}$, respectively. The shadows and error bars show one standard deviation.

Figure S1 presents the regret of the two versions of our DEL algorithm. Clearly, exploiting the structure brings a very significant performance improvement and the gain grows as the number of states increases, as predicted by our theoretical results. Observe that the regret after $T = 50k$ steps under the version of DEL exploiting the Lipschitz structure barely grows with the number of states, see Figure S1(b), which was also expected.

## C   Proof of Theorem 1

**Notations and preliminaries.** Let $N_T(x) = \sum_{t=1}^T \mathbb{1}[X_t = x]$ and $N_T(x,a) = \sum_{t=1}^T \mathbb{1}[X_t = x, A_t = a]$ denote the number of times $x$ and $(x,a)$ have been visited up to step $T$. For any $\psi \in \Phi$ and any initial state $x_1$, we denote by $\mathbb{P}_{\psi|x_1}^\pi$ and $\mathbb{E}_{\psi|x_1}^\pi$ the probability measure and expectation under $\pi$ and $\psi$ conditioned on $X_1 = x_1$. The regret up to step $T$ starting in state $x_1$ under $\pi$ and $\psi$ is denoted by $R_{T,\psi}^\pi(x_1)$. To emphasize the dependence on the MDP $\psi$ of the gap function $\delta^*$, we further denote its value at $(x,a)$ by $\delta^*(x,a;\psi)$.

For any $\psi \in \Phi$, using the ergodicity of $\psi$, we may leverage the same arguments as those used in the proof of Proposition 1 of [Burnetas and Katehakis, 1997] to establish a connection between the regret of an algorithm $\pi \in \Pi$ under $\psi$ and $N_T(x,a)$. Specifically, for any $x_1$,

$$R_{T,\psi}^\pi(x_1) = \sum_{x \in \mathcal{S}} \sum_{a \notin \mathcal{O}(y,\psi)} \mathbb{E}_{\psi|x_1}^\pi[N_T(x,a)]\delta^*(x,a;\psi) + O(1), \quad \text{as } T \to \infty. \tag{S3}$$

In addition, due to the ergodicity of $\psi$, we can also prove as in Proposition 2 in [Burnetas and Katehakis, 1997] that there exists constants $C, \rho > 0$ such that for any $x \in \mathcal{S}, \pi \in \Pi$,

$$\mathbb{P}_{\psi|x}^\pi[N_T(x) \leq \rho T] \leq C \cdot \exp(-\rho T/2). \tag{S4}$$

**Change-of-measure argument.** Let $\pi$ be a uniformly good algorithm, and $x_1$ an initial state. For any *bad* MDP $\psi \in \Delta_\Phi(\phi)$, the argument consists in (i) relating the log-likelihood of the observations under $\phi$ and $\psi$ to the expected number of times sub-optimal actions are selected under $\pi$, and (ii) using the fact that $\pi$ is uniformly good to derive a lower bound on the log-likelihood.

(i) Define by $L$ the log-likelihood of the observations up to step $T$ under $\phi$ and $\psi$. We can use the same techniques as in [Kaufmann et al., 2016, Garivier et al., Jun. 2018] (essentially an extension of Wald's lemma):

$$\mathbb{E}^{\pi}_{\phi|x_1}[L] = \sum_{x,a} \mathbb{E}^{\pi}_{\phi|x_1}[N_T(x,a)]\text{KL}_{\phi|\psi}(x,a). \tag{S5}$$

The so-called data processing inequality [Garivier et al., Jun. 2018] yields for all event $\mathcal{E}$ in $\mathcal{H}^{\pi}_T$:
$\mathbb{E}^{\pi}_{\phi|x_1}[L] \geq kl(\mathbb{P}^{\pi}_{\phi|x_1}[\mathcal{E}], \mathbb{P}^{\pi}_{\psi|x_1}[\mathcal{E}])$, where for $u, v \in [0, 1]$, $kl(u, v) := u \log \frac{u}{v} + (1-u) \log \frac{1-u}{1-v}$.
Combine with (S5), this leads to:

$$\sum_{x,a \notin \mathcal{O}(x,\phi)} \mathbb{E}^{\pi}_{\phi|x_1}[N_T(x,a)]\text{KL}_{\phi|\psi}(x,a) \geq \text{KL}(\mathbb{P}^{\pi}_{\phi|x_1}[\mathcal{E}], \mathbb{P}^{\pi}_{\psi|x_1}[\mathcal{E}]). \tag{S6}$$

Note that in the above sum, we removed $a \in \mathcal{O}(x,\phi)$ since $\text{KL}_{\phi|\psi}(x,a) = 0$ if $a \in \mathcal{O}(x,\phi)$.

(ii) Next we will leverage the fact that $\pi$ is uniformly good to select the event $E$. We first state the following lemma, proved at the end of this section. Since $\Pi^*(\phi) \cap \Pi^*(\psi) = \emptyset$, there exists $x \in \mathcal{S}$ such that for all $\alpha > 0$,

$$\mathbb{E}^{\pi}_{\psi|x_1}\left[\sum_{a \in \mathcal{O}(x,\phi)} N_T(x,a)\right] = o(T^{\alpha}).$$

Indeed, otherwise $\pi$ would not be uniformly good. Now define the event $\mathcal{E}$ as:

$$\mathcal{E} := \left[N_T(x) \geq \rho T, \sum_{a \notin \mathcal{O}(x,\phi)} N_T(x,a) \leq \sqrt{T}\right],$$

where the constant $\rho$ is chosen so that (S4) holds under $\phi$ and $\psi$. Using a union bound, we have

$$1 - \mathbb{P}^{\pi}_{\phi|x_1}[\mathcal{E}] \leq \mathbb{P}^{\pi}_{\phi|x_1}[N_T(x) \leq \rho T] + \mathbb{P}^{\pi}_{\phi|x_1}\left[\sum_{a \notin \mathcal{O}(x,\phi)} N_T(x,a) \geq \sqrt{T}\right]$$

$$\leq C \cdot \exp(-\rho T/2) + \frac{\mathbb{E}^{\pi}_{\phi|x_1}\left[\sum_{a \notin \mathcal{O}(x,\phi)} N_T(x,a)\right]}{\sqrt{T}} \tag{S7}$$

where for the first and second terms in the last inequality, we used (S4) and Markov inequality, respectively. Since $\pi$ is uniformly good, $\mathbb{E}^{\pi}_{\phi|x_1}[\sum_{a \notin \mathcal{O}(x,\phi)} N_T(x,a)] = o(T^{\alpha})$ for all $\alpha > 0$, the last term of (S7) converges to 0, i.e., $\mathbb{P}^{\pi}_{\phi|x_1}[\mathcal{E}] \to 1$ as $T \to \infty$. Using Markov inequality, it follows that

$$\mathbb{P}^{\pi}_{\psi|x_1}[\mathcal{E}] \leq \mathbb{P}^{\pi}_{\psi|x_1}\left[N_T(x) - \sum_{a \notin \mathcal{O}(x,\phi)} N_T(x,a) \geq \rho T - \sqrt{T}\right]$$

$$\leq \frac{\mathbb{E}^{\pi}_{\psi|x_1}[N_T(x) - \sum_{a \notin \mathcal{O}(x,\phi)} N_T(x,a)]}{\rho T - \sqrt{T}} = \frac{\mathbb{E}^{\pi}_{\psi|x_1}[\sum_{a \in \mathcal{O}(x,\phi)} N_T(x,a)]}{\rho T - \sqrt{T}}$$

which converges to 0 because of our choice of $x$. Combining $\mathbb{P}^{\pi}_{\phi|x_1}[\mathcal{E}] \to 1$ and $\mathbb{P}^{\pi}_{\psi|x_1}[\mathcal{E}] \to 0$,

$$\frac{kl(\mathbb{P}^{\pi}_{\phi|x_1}[\mathcal{E}], \mathbb{P}^{\pi}_{\psi|x_1}[\mathcal{E}])}{\log T} \underset{T \to \infty}{\sim} \frac{1}{\log T} \log\left(\frac{1}{\mathbb{P}^{\pi}_{\psi|x_1}[\mathcal{E}]}\right) \geq \frac{1}{\log T} \log\left(\frac{\rho T - \sqrt{T}}{\mathbb{E}^{\pi}_{\psi|x_1}[\sum_{a \in \mathcal{O}(x,\phi)} N_T(x,a)]}\right)$$

which converges to 1 as $T$ grows large due to our choice of $x$. Plugging this result in (S6), we get:

$$\liminf_{T \to \infty} \frac{1}{\log T} \sum_{x,a \notin \mathcal{O}(x,\phi)} \mathbb{E}^{\pi}_{\phi|x_1}[N_T(x,a)]\text{KL}_{\phi|\psi}(x,a) \geq 1. \tag{S8}$$

Combining the above constraints valid for any $\psi \in \Delta_{\Phi}(\psi)$ and (S3) concludes the proof of the theorem.

# D    Proof of Theorem 2

We first prove the decoupling lemma.

**Proof of Lemma 1**. We prove the lemma by contradiction. Assume that $\Pi^*(\phi) \cap \Pi^*(\psi_1) \neq \emptyset$ and $\Pi^*(\phi) \cap \Pi^*(\psi_2) \neq \emptyset$. Let $\Pi^*(\phi, \psi_1, \psi_2) := \Pi^*(\phi) \cap \Pi^*(\psi_1) \cap \Pi^*(\psi_2)$. It is sufficient to show

$$(i) \; \Pi^*(\phi, \psi_1, \psi_2) \neq \emptyset \,, \quad \text{and} \quad (ii) \; \Pi^*(\phi, \psi_1, \psi_2) \subseteq \Pi^*(\psi_0) \,. \tag{S9}$$

Indeed, this implies $\Pi^*(\phi) \cap \Pi^*(\psi_0) \neq \emptyset$. Note that any policy $f \in \Pi^*(\phi)$ has the same gain and bias function under $\phi, \psi_0, \psi_1, \psi_2$ since the modifications of $\phi$ are made on suboptimal (state, action) pairs. Specifically,

$$g_{\psi_0}^f = g_{\psi_1}^f = g_{\psi_2}^f = g_\phi^* \quad \text{and} \quad h_{\psi_0}^f = h_{\psi_1}^f = h_{\psi_2}^f = h_\phi^* \,. \tag{S10}$$

To prove (i), the first part of (S9), consider a policy $f' \in \Pi^*(\phi) \cap \Pi^*(\psi_1)$ and a policy $f'' \in \Pi^*(\phi) \cap \Pi^*(\psi_2)$. Then, from the optimality of $f'$ under $\psi_1$, it follows that for each $x \in \mathcal{S}$,

$$g_{\psi_1}^* = g_{\psi_1}^{f'} \geq g_{\psi_1}^{f''} = g_{\psi_2}^{f''} = g_{\psi_2}^* \,, \tag{S11}$$

where for the second equality, we use (S10). Similarly, we have for each $x \in \mathcal{S}$,

$$g_{\psi_2}^* = g_{\psi_2}^{f''} \geq g_{\psi_2}^{f'} = g_{\psi_1}^{f'} = g_{\psi_1}^* \,.$$

Hence $g_{\psi_1}^* = g_{\psi_2}^*$ and $\Pi^*(\phi, \psi_1, \psi_2) = \Pi^*(\phi) \cap \Pi^*(\psi_1) = \Pi^*(\phi) \cap \Pi^*(\psi_2) \neq \emptyset$.

We now prove (ii), the second part of (S9). Let $f \in \Pi^*(\phi, \psi_1, \psi_2)$. It is sufficient to show $g_{\psi_0}^f$ and $h_{\psi_0}^f(x)$ verify the Bellman optimality equation for model $\psi_0$. Using (S10) and the optimality of $f$ under $\psi_1$, for all $a \in \mathcal{A}$, if $(x, a) \notin \mathcal{U}_2$,

$$r_{\psi_0}(x, f(x)) + \sum_{y \in \mathcal{S}} p_{\psi_0}(y|x, f(x)) h_{\psi_0}^f(y) \overset{(a)}{=} r_{\psi_1}(x, f(x)) + \sum_{y \in \mathcal{S}} p_{\psi_1}(y|x, f(x)) h_{\psi_1}^f(y)$$

$$\overset{(b)}{\geq} r_{\psi_1}(x, a) + \sum_{y \in \mathcal{S}} p_{\psi_1}(y|x, a) h_{\psi_1}^f(y)$$

$$\overset{(c)}{=} r_{\psi_0}(x, a) + \sum_{y \in \mathcal{S}} p_{\psi_0}(y|x, a) h_{\psi_0}^f(y) \,, \tag{S12}$$

where for (a) and (c), we used (S10) and the fact that the kernels of $\psi_0$ and $\psi_1$ are the same at every $(x, a) \notin \mathcal{U}_2$, and for (b), we used the fact that $g_{\psi_1}^f$ and $h_{\psi_1}^f$ verify the Bellman optimality equation for $\psi_1$. Similarly, using the optimality of $f$ under $\psi_2$, it follows that for $(x, a) \in \mathcal{U}_2$,

$$r_{\psi_0}(x, f(x)) + \sum_{y \in \mathcal{S}} p_{\psi_0}(y|x, f(x)) h_{\psi_0}^f(y) = r_{\psi_2}(x, f(x)) + \sum_{y \in \mathcal{S}} p_{\psi_2}(y|x, f(x)) h_{\psi_2}^f(y)$$

$$\geq r_{\psi_2}(x, a) + \sum_{y \in \mathcal{S}} p_{\psi_2}(y|x, a) h_{\psi_2}^f(y)$$

$$= r_{\psi_0}(x, a) + \sum_{y \in \mathcal{S}} p_{\psi_0}(y|x, a) h_{\psi_0}^f(y) \,. \tag{S13}$$

Combining (S12) and (S13), for all $(x, a) \in \mathcal{S} \times \mathcal{A}$,

$$r_{\psi_0}(x, f(x)) + \sum_{y \in \mathcal{S}} p_{\psi_0}(y|x, f(x)) h_{\psi_0}^f(y) \geq r_{\psi_0}(x, a) + \sum_{y \in \mathcal{S}} p_{\psi_0}(y|x, a) h_{\psi_0}^f(y) \,,$$

which implies that $g_{\psi_0}^f$ and $h_{\psi_0}^f$ verify the Bellman optimality equation under model $\psi_0$, i.e., $f \in \Pi^*(\psi_0)$. $\qquad\square$

**Proof of Theorem 2**. Recall that any policy $f \in \Pi^*(\phi)$ has the same gain and bias function in $\psi$ and $\phi$ since the kernels of $\phi$ and $\psi$ are identical at every $(x, a)$ such that $a \in \mathcal{O}(x; \phi)$. More formally, for any $f \in \Pi^*(\phi)$,

$$\mathbf{B}_\phi^f = \mathbf{B}_\psi^f, \quad g_\phi^* = g_\phi^f = g_\psi^f \quad \text{and} \quad h_\phi^*(\cdot) = h_\phi^f(\cdot) = h_\psi^f(\cdot) \,.$$

Next we show that for all $\psi \in \Delta_\Phi(\phi)$,

$$\Pi^*(\phi) \cap \Pi^*(\psi) = \emptyset \implies \exists (x, a) \text{ such that } (\mathbf{B}^a_\psi h^*_\phi)(x) > g^*_\phi + h^*_\phi(x) . \tag{S14}$$

We prove (S14) by contradiction. Consider a policy $f \in \Pi^*(\phi)$. Suppose that for all $(x, a)$, $(\mathbf{B}^a_\psi h^f_\phi)(x) \le g^f_\phi + h^f_\phi(x)$. Then, for all $(x, a)$,

$$(\mathbf{B}^f_\psi h^f_\psi)(x) = (\mathbf{B}^f_\phi h^f_\phi)(x) = g^f_\phi + h^f_\phi(x) \ge \max_{a \in \mathcal{A}}(\mathbf{B}^a_\psi h^f_\phi)(x)$$

which implies that $g^f_\psi$ and $h^f_\psi$ verify the Bellman optimality equation under $\psi$. Hence, $f \in \Pi^*(\psi)$ which contradicts to $\Pi^*(\phi) \cap \Pi^*(\psi) = \emptyset$.

Finally Theorem 2 is obtained by combining the decoupling lemma and (S14). Indeed, due to the decoupling lemma, we may restrict $\Delta_\Phi(\phi)$ to MDPs obtained from $\phi$ by only changing the kernels in a single state-action pair. $\qquad \square$

**Simplification for null structure**. We conclude this section by proving that $\mathcal{F}_{\mathrm{un}}(\phi) \subset \mathcal{F}_\Phi(\phi)$. Let $\eta \in \mathcal{F}_{\mathrm{un}}(\phi)$, recalling that

$$\mathcal{F}_{\mathrm{un}}(\phi) = \left\{ \eta \in \mathcal{F}_0(\phi) : \eta(x, a) \left( \frac{\delta^*(x, a; \phi)}{H+1} \right)^2 \ge 2, \ \forall (x, a) \text{ s.t. } a \notin \mathcal{O}(x; \phi) \right\} .$$

We show that $\eta \in \mathcal{F}_\Phi(\phi)$. To this aim, we need to show that $\forall (x, a)$ s.t. $a \notin \mathcal{O}(x; \phi)$,

$$\eta(x, a)\mathrm{KL}_{\phi|\psi}(x, a) \ge 1, \ \forall \psi \in \Delta_\Phi(x, a; \phi).$$

Let $(x, a)$ be such that $a \notin \mathcal{O}(x; \phi)$, which means $a \notin \mathcal{O}(x, h^*_\phi; \phi)$, and $\psi \in \Delta_\Phi(x, a; \phi)$. We have, by definition, $(\mathbf{B}^a_\psi h^*_\phi)(x) > g^*_\phi + h^*_\phi(x)$. Then,

$$\begin{aligned}
\delta^*(x, a; \phi) &= (\mathbf{B}^*_\phi h^*_\phi)(x) - (\mathbf{B}^a_\phi h^*_\phi)(x) \\
&< (\mathbf{B}^a_\psi h^*_\phi)(x) - (\mathbf{B}^a_\phi h^*_\phi)(x) \\
&= r_\psi(x, a) - r_\phi(x, a) + \sum_{y \in \mathcal{S}}(p_\psi(y \mid x, a) - p_\phi(y \mid x, a))h^*_\phi(y) \\
&\le \|q_\psi(\cdot \mid x, a) - q_\phi(\cdot \mid x, a)\|_1 + H\|p_\psi(\cdot \mid x, a) - p_\phi(\cdot \mid x, a)\|_1 \\
&\le (H+1)\|\psi(x, a) - \phi(x, a)\|_1
\end{aligned}$$

where we define

$$\|\psi(x, a) - \phi(x, a)\|_1 := \|q_\psi(\cdot \mid x, a) - q_\phi(\cdot \mid x, a)\|_1 + \|p_\psi(\cdot \mid x, a) - p_\phi(\cdot \mid x, a)\|_1.$$

Finally, Pinsker's inequality yields:

$$2\mathrm{KL}_{\phi|\psi}(x, a) \ge \|\psi(x, a) - \phi(x, a)\|_1^2 \ge \left( \frac{\delta^*(x, a; \phi)}{H+1} \right)^2 .$$

This implies that:

$$\eta(x, a)\mathrm{KL}_{\phi|\psi}(x, a) \ge \frac{\eta(x, a)}{2} \left( \frac{\delta^*(x, a; \phi)}{H+1} \right)^2 \ge 1$$

where the last inequality is due to the fact that $\eta(x, a) \in \mathcal{F}_{\mathrm{un}}(\phi)$.

# E   Proof of Theorem 3

We prove that $\mathcal{F}_{\text{lip}}(\phi) \subset \mathcal{F}_\Phi(\phi)$. Let $\eta \in \mathcal{F}_{\text{lip}}(\phi)$. We show that $\eta \in \mathcal{F}_\Phi(\phi)$. Let $\psi \in \Delta_\Phi(\phi)$, then, from (S14), there exist $(x', a')$ such that $(\mathbf{B}_\psi^{a'} h_\phi^*)(x') > g_\phi^* + h_\phi^*(x')$. Then, using the same arguments as at the end of the previous section, we obtain:

$$\|\phi(x', a') - \psi(x', a')\|_1 \geq \frac{\delta^*(x', a'; \phi)}{H + 1}. \tag{S15}$$

Now for all $(x, a) \in \mathcal{S} \times \mathcal{A}$,

$$\|\phi(x', a') - \psi(x', a')\|_1 \leq \|\phi(x', a') - \phi(x, a)\|_1 + \|\phi(x, a) - \psi(x, a)\|_1 + \|\psi(x, a) - \psi(x', a')\|_1$$
$$\leq \|\phi(x, a) - \psi(x, a)\|_1 + 2Ld(x, x')^\alpha + 2L'd(a, a')^{\alpha'}, \tag{S16}$$

where the first inequality follows from the triangular inequality and the second follows from Lipschitz continuity. This further implies that

$$\|\phi(x, a) - \psi(x, a)\|_1 \geq \left[\frac{\delta^*(x', a'; \phi)}{H + 1} - 2\Big(Ld(x, x')^\alpha + L'd(a, a')^{\alpha'}\Big)\right]_+.$$

Hence, using Pinsker's inequality,

$$2\text{KL}_{\phi|\psi}(x, a) \geq \left[\frac{\delta^*(x', a'; \phi)}{H + 1} - 2\Big(Ld(x, x')^\alpha + L'd(a, a')^{\alpha'}\Big)\right]_+^2, \tag{S17}$$

which implies that:

$$\eta(x, a)\text{KL}_{\phi|\psi}(x, a) \geq \frac{\eta(x, a)}{2} \left[\frac{\delta^*(x', a'; \phi)}{H + 1} - 2\Big(Ld(x, x')^\alpha + L'd(a, a')^{\alpha'}\Big)\right]_+^2 \geq 1. \tag{S18}$$

The last inequality follows from $\eta \in \mathcal{F}_{\text{lip}}$. Thus $\mathcal{F}_{\text{lip}}(\phi) \subset \mathcal{F}_\Phi(\phi)$.

Next we derive an upper bound for $K_\Phi(\phi)$. To this aim, we construct a vector $\eta \geq 0$ verifying (2b) for our given structure $\Phi$. Then, we get an upper bound of $K_\Phi(\phi)$ by evaluating the objective function of $P(\phi, \mathcal{F}_\Phi(\phi))$ at $\eta$.

To construct $\eta$, we build a sequence $(\mathcal{X}_i)_{i=1,2,\ldots}$ of sets of (state, action) pairs, as well as a sequence $(x_i)_{i=1,2,\ldots}$(state, action) pairs, such that for any $i \geq 1$, $\mathcal{X}_{i+1} \subset \mathcal{X}_i$, and $(x_i, a_i) \in \arg\max_{(x,a)\in\mathcal{X}_i} \delta^*(x, a; \phi)$ (ties are broken arbitrarily).

We start with $\mathcal{X}_1 = \{(x, a) : x \in \mathcal{S}, a \notin \mathcal{O}(x; \phi), i.e., \delta^*(x, a; \phi) > 0\}$. Recursively, for each $i = 1, 2, \ldots$, let

$$\mathcal{B}_i = \left\{(x, a) \in \mathcal{X}_i : Ld(x, x_i)^\alpha + L'd(a, a_i)^{\alpha'} \leq \frac{\delta_{\min}}{4(H + 1)}\right\}, \quad \text{and}$$
$$\mathcal{X}_{i+1} = \mathcal{X}_i \setminus \mathcal{B}_i . \tag{S19}$$

Let $I$ be the first index such that $\mathcal{X}_{I+1} = \emptyset$. Construct $\eta$ as

$$\eta(x, a) = \begin{cases} 8\left(\frac{\delta_{\min}}{H+1}\right)^{-2} & \text{if } \exists i \in [1, I] \text{ such that } (x, a) = (x_i, a_i), \\ 0 & \text{otherwise.} \end{cases} \tag{S20}$$

Observe that $\eta$ is strictly positive at only $I$ pairs, and hence

$$\sum_{(x,a)\in\mathcal{S}\times\mathcal{A}} \delta^*(x, a; \phi)\eta(x, a) \leq 8(H + 1)\left(\frac{H + 1}{\delta_{\min}}\right)^2 I$$

since $\delta^*(x, a; \phi) \leq H + 1$ for all $(x, a)$. Next, we bound $I$ using the covering and packing numbers of the hypercubes $[0, D]^d$ and $[0, D']^{d'}$.

**Lemma S1.** *The generation of $\mathcal{X}_i$'s in* (S19) *must stop after* $(S_{\text{lip}}A_{\text{lip}}+1)$ *iterations, i.e.,* $I \leq S_{\text{lip}}A_{\text{lip}}$.

The proof of this lemma is postponed at the end of this section. To complete the proof of the theorem, it remains to show that $\eta$ verifies all the constraints (2b) for the Lipschitz structure $\Phi$.

Remember that $\mathcal{F}_{\text{lip}}(\phi) \subset \mathcal{F}_{\Phi}(\phi)$. Fix $\psi \in \Delta_{\Phi}(\phi)$. There exists $(x', a')$ such that $a' \notin \mathcal{O}(x'; h_{\phi}^*, \phi)$ and $a' \in \mathcal{O}(x'; h_{\phi}^*, \psi)$, and such that (S15) holds. Let $i \in \{1, \ldots, I\}$ denote an index such that $(x', a') \in \mathcal{B}_i$. Note that such an index $i$ exists since $(x', a') \in \mathcal{X}_1$ and $\mathcal{X}_{I+1} = \emptyset$. Thus, we have:

$$
\begin{aligned}
\sum_{(x,a)\in\mathcal{S}\times\mathcal{A}} \eta(x,a)\text{KL}_{\phi|\psi}(x,a) &\geq \sum_{(x,a)\in\mathcal{S}\times\mathcal{A}} \frac{\eta(x,a)}{2}\left[\frac{\delta^*(x',a';\phi)}{H+1} - 2\Big(Ld(x,x')^{\alpha} + L'd(a,a')^{\alpha'}\Big)\right]_+^2 \\
&\geq \frac{\eta(x_i,a_i)}{2}\left[\frac{\delta^*(x',a';\phi)}{H+1} - 2\Big(Ld(x_i,x')^{\alpha} + L'd(a_i,a')^{\alpha'}\Big)\right]_+^2 \\
&\geq \frac{\eta(x_i,a_i)}{2}\left[\frac{\delta^*(x',a';\phi)}{H+1} - \frac{1}{2}\frac{\delta_{\min}}{H+1}\right]_+^2 \\
&\geq 4\left(\frac{\delta_{\min}}{H+1}\right)^{-2}\left(\frac{1}{2}\frac{\delta_{\min}}{H+1}\right)^2 = 1
\end{aligned}
$$

where the third inequality follows from the fact that $(x', a') \in \mathcal{B}_i$. Hence we have verified that $\eta$ satisfies the feasibility constraint for $\psi$. Since this observation holds for all $\psi \in \Delta_{\Phi}(\phi)$, this completes the proof of Theorem 3. $\qquad\square$

**Proof of Lemma S1.** A $\delta$-*packing* of a set $\mathcal{D}$ with respect to a metric $\rho$ is a set $\{x_1, \ldots, x_n\} \subset \mathcal{D}$ such that $\rho(x_i - x_j) > \delta$ for all different $i, j \in \{1, \ldots, n\}$. The $\delta$-*packing number* $I_{\text{p}}(\delta, \mathcal{D}, \rho)$ is the cardinality of the largest $\delta$-packing. The construction of $\mathcal{X}_i$ ensures that for different $i, j \in \{1, \ldots, I\}$,

$$
\ell_{\text{lip}}((x_i, a_i), (x_j, a_j)) > \delta := \frac{\delta_{\min}}{4(H+1)} \ ,
$$

where for $(x, a), (x', a') \in \mathbb{R}^d \times \mathbb{R}^{d'}$,

$$
\ell_{\text{lip}}((x, a), (x', a')) := Ld(x, x')^{\alpha} + L'd(a, a')^{\alpha'} \ .
$$

Then, we have:

$$
I \leq I_{\text{p}}\left(\delta, \mathcal{S} \times \mathcal{A}, \ell_{\text{lip}}\right) \ .
$$

To obtain an upper bound of the packing number, we further define the covering number. A $\delta$-cover of a set $\mathcal{D}$ with respect to a metric $\rho$ is a set $\{x_1, \ldots, x_I\} \subset \mathcal{D}$ such that for each $x \in \mathcal{D}$, there exists some $i \in \{1, \ldots, I\}$ such that $\rho(x, x_i) \leq \delta$. The $\delta$-covering number $I_{\text{c}}(\delta, \mathcal{D}, \rho)$ is the smallest cardinality of $\delta$-cover. Then, we have the following relationship between the packing and covering numbers.

**Lemma S2.** *For all $\delta > 0$, $\mathcal{D}, \mathcal{D}'$ such that $\mathcal{D} \subset \mathcal{D}'$,*

$$
I_{\text{p}}(2\delta, \mathcal{D}, \rho) \leq I_{\text{c}}(\delta, \mathcal{D}, \rho) \leq I_{\text{c}}(\delta, \mathcal{D}', \rho) \ .
$$

The proof of this lemma is provided at the end of the section for completeness. Define the metrics $\ell_{\max}^{(1)}, \ell_{\max}^{(2)}, \ell_{\max}$ for $\mathbb{R}^d, \mathbb{R}^{d'}, \mathbb{R}^d \times \mathbb{R}^{d'}$, respectively, as follows:

$$
\ell_{\max}^{(1)}(x, x') := \left(\frac{1}{\sqrt{d}}\left(\frac{\delta}{2L}\right)^{1/\alpha}\right)^{-1}\|x - x'\|_{\infty} \ ,
$$

$$
\ell_{\max}^{(2)}(a, a') := \left(\frac{1}{\sqrt{d'}}\left(\frac{\delta}{2L'}\right)^{1/\alpha'}\right)^{-1}\|a - a'\|_{\infty} \ ,
$$

$$
\ell_{\max}((x, a), (x', a')) := \max\left\{\ell_{\max}^{(1)}(x, x'), \ell_{\max}^{(2)}(a, a')\right\} \ ,
$$

where $\|\cdot\|_{\infty}$ is infinite norm. Then, it follows that for any $(x, a), (x', a') \in \mathbb{R}^d \times \mathbb{R}^{d'}$,

$$
\ell_{\max}((x, a), (x', a')) \leq 1 \implies \ell_{\text{lip}}((x, a), (x', a')) \leq \delta \ .
$$

Hence, we have

$$I \leq I_{\mathrm{c}}(\delta, \mathcal{S} \times \mathcal{A}, \ell_{\mathrm{lip}})$$
$$\leq I_{\mathrm{c}}(1, \mathcal{S} \times \mathcal{A}, \ell_{\mathrm{max}})$$
$$\leq I_{\mathrm{c}}(1, \mathcal{S}, \ell_{\mathrm{max}}^{(1)}) I_{\mathrm{c}}(1, \mathcal{A}, \ell_{\mathrm{max}}^{(2)})$$

since for any 1-cover $\mathcal{S}'$ of $\mathcal{S}$ with metric $\ell_{\mathrm{max}}^{(1)}$ and any 1-cover $\mathcal{A}'$ of $\mathcal{A}$ with metric $\ell_{\mathrm{max}}^{(2)}$, their Cartesian product $\mathcal{S}' \times \mathcal{A}' = \{(x, a) : x \in \mathcal{S}', a \in \mathcal{A}'\}$ is 1-cover of $\mathcal{S} \times \mathcal{A}$ with metric $\ell_{\mathrm{max}}$. We now study $I_{\mathrm{c}}(1, \mathcal{S}, \ell_{\mathrm{max}}^{(1)})$ and $I_{\mathrm{c}}(1, \mathcal{A}, \ell_{\mathrm{max}}^{(2)})$. Recalling $\mathcal{S} \subset [0, D]^d$ and using Lemma S2, it follows directly that

$$I_{\mathrm{c}}(1, \mathcal{S}, \ell_{\mathrm{max}}^{(1)}) \leq I_{\mathrm{c}}(1, [0, D]^d, \ell_{\mathrm{max}}^{(1)})$$
$$= I_{\mathrm{c}}\left(\frac{1}{\sqrt{d}}\left(\frac{\delta}{2L}\right)^{1/\alpha}, [0, D]^d, \|\cdot\|_\infty\right)$$
$$\leq \left(\frac{D}{\frac{1}{\sqrt{d}}\left(\frac{\delta}{2L}\right)^{1/\alpha}} + 1\right)^d,$$

which implies

$$I_{\mathrm{c}}(1, \mathcal{S}, \ell_{\mathrm{max}}^{(1)}) \leq \min\left\{|\mathcal{S}|, \left(\frac{D}{\frac{1}{\sqrt{d}}\left(\frac{\delta}{2L}\right)^{1/\alpha}} + 1\right)^d\right\} = S_{\mathrm{lip}}$$

where we used the fact that $I_{\mathrm{c}}(1, \mathcal{S}, \ell_{\mathrm{max}}^{(1)}) \leq |\mathcal{S}|$. Similarly, we have

$$I_{\mathrm{c}}(1, \mathcal{A}, \ell_{\mathrm{max}}^{(2)}) \leq \min\left\{|\mathcal{A}|, \left(\frac{D'}{\frac{1}{\sqrt{d'}}\left(\frac{\delta}{2L'}\right)^{1/\alpha'}} + 1\right)^{d'}\right\} = A_{\mathrm{lip}}.$$

This completes the proof of Lemma S1. $\qquad\square$

**Proof of Lemma S2.** Consider a $\delta$-cover $\mathcal{X}$ and a $2\delta$-packing $\mathcal{Y}$ of set $\mathcal{D}$ with respect to metric $\rho$. Then, there is no $x \in \mathcal{X}$ such that $y, y' \in \mathcal{B}(\delta, x) = \{x' \in \mathcal{D} : \rho(x, x') \leq \delta\}$ for two different $y, y' \in \mathcal{Y}$. Otherwise, we would have $\rho(x, y) \leq \delta$ and $\rho(x, y') \leq \delta$ which implies $\rho(y, y') \leq 2\delta$ from the triangle inequality, and contradicts the fact that $y, y'$ are two different elements of $2\delta$-cover, i.e., $\rho(y, y') > 2\delta$. Thus, the cardinality of $\mathcal{Y}$ cannot be larger than that of $\mathcal{X}$. Due to the arbitrary choice of $\delta$-cover $\mathcal{X}$ and a $2\delta$-packing $\mathcal{Y}$, we conclude that $I_{\mathrm{p}}(2\delta, \mathcal{D}, \rho) \leq I_{\mathrm{c}}(\delta, \mathcal{D}, \rho)$.

The second inequality in the lemma is straightforward. $\qquad\square$

# F    Proof of Theorem 4

We analyze the regret under $\pi = \mathrm{DEL}$ algorithm when implemented with the original feasible set $\mathcal{F}_\Phi(\phi; \mathcal{C}, \zeta)$. Extending the analysis to the case where DEL runs on the simplified feasible sets $\mathcal{F}_{\mathrm{un}}(\phi; \mathcal{C}, \zeta)$ and $\mathcal{F}_{\mathrm{lip}}(\phi; \mathcal{C}, \zeta)$ can be easily done.

For $T \geq 1$, $\varepsilon > 0$, $x \in \mathcal{S}$ and $a \in \mathcal{A}$, define the following random variables:

$$W_T^{(1)}(x, a; \varepsilon) := \sum_{t=1}^T \mathbb{1}\left[(X_t, A_t) = (x, a), \mathcal{E}_t(\varepsilon), (\mathbf{B}_{\phi_t}^a h_t')(x) \leq (\mathbf{B}_\phi^* h_\phi^*)(x) - 2\varepsilon\right]$$

$$W_T^{(2)}(x, a; \varepsilon) := \sum_{t=1}^T \mathbb{1}\left[(X_t, A_t) = (x, a), \mathcal{E}_t(\varepsilon), (\mathbf{B}_{\phi_t}^a h_t')(x) > (\mathbf{B}_\phi^* h_\phi^*)(x) - 2\varepsilon\right]$$

$$W_T^{(3)}(\varepsilon) := \sum_{t=1}^T \mathbb{1}\left[\neg\mathcal{E}_t(\varepsilon)\right]$$

where we use the standard notation $\neg\mathcal{U}$ to represent the event that $\mathcal{U}$ does not occur, where we recall that $h'_t := h_{\phi'_t}$ is the bias function of the restricted estimated model $\phi'_t = \phi_t(\mathcal{C}_t)$ at time $t$, and where the event $\mathcal{E}_t(\varepsilon)$ is defined as:

$$\mathcal{E}_t(\varepsilon) := \left\{\Pi^*(\phi'_t) \subseteq \Pi^*(\phi) \text{ and } |r_t(x,a) - r_\phi(x,a)| + |h'_t(x) - h^*_\phi(x)| \le \varepsilon \; \forall x \in \mathcal{S}, \forall a \in \mathcal{O}(x;\phi'_t)\right\} \;.$$

From the above definitions, we have:

$$R^\pi_T(x_1) \le \sum_{(x,a):a\notin\mathcal{O}(x;\phi)} \delta^*(x,a;\phi)\, \mathbb{E}^\pi_{\phi|x_1}\left[W^{(1)}_T(x,a;\varepsilon)\right] \tag{S21a}$$

$$+ \sum_{(x,a):a\notin\mathcal{O}(x;\phi)} S\, \mathbb{E}^\pi_{\phi|x_1}\left[W^{(2)}_T(x,a;\varepsilon)\right] \tag{S21b}$$

$$+ S\, \mathbb{E}^\pi_{\phi|x_1}[W^{(3)}_T(\varepsilon)] \;. \tag{S21c}$$

The multiplicative factor $S$ in the last two terms arises from the fact that $\max_{(x,a)} \delta^*(x,a;\phi) \le S$ when the magnitude of the instantaneous reward is bounded by 1. Next we provide upper bounds of each of the three terms in (S21).

**A. Upper bounds for (S21a) and (S21b).** To study the first two terms in (S21), we first make the following observations on the behavior of the algorithm. Let $\mathcal{E}^{\mathrm{est}}_t$, $\mathcal{E}^{\mathrm{mnt}}_t$, $\mathcal{E}^{\mathrm{xpt}}_t$, and $\mathcal{E}^{\mathrm{xpr}}_t$ denote the events that at time $t$, the algorithm enters the estimation, monotonization, exploitation, and exploration phases, respectively. By the design of the algorithm, the estimation phase generates regret no more than $O(\log T / \log\log T) = o(\log T)$, i.e.,

$$\sum_{t=1}^T \mathbb{1}[\mathcal{E}^{\mathrm{est}}_t] = o(\log T). \tag{S22}$$

Moreover, when the event $\mathcal{E}_t(\varepsilon)$ occurs, we have $\mathcal{O}(X_t;\phi'_t) \subseteq \mathcal{O}(X_t;\phi)$ and thus the monotonization and exploitation phases produce no regret. Formally, for $(x,a) \in \mathcal{S} \times \mathcal{A}$ such that $a \notin \mathcal{O}(x;\phi)$,

$$\sum_{t=1}^T \mathbb{1}[(X_t,A_t) = (x,a), \mathcal{E}_t(\varepsilon), \mathcal{E}^{\mathrm{mnt}}_t \cup \mathcal{E}^{\mathrm{xpt}}_t] = 0. \tag{S23}$$

Hence, when $\mathcal{E}_t(\varepsilon)$ occurs, we just care about the regret generated in the exploration phase, i.e., for any $(x,a) \in \mathcal{S} \times \mathcal{A}$ such that $a \notin \mathcal{O}(x;\phi)$,

$$\mathbb{E}^\pi_{\phi|x_1}\left[W^{(1)}_T(x,a;\varepsilon)\right] \le o(\log T) + \sum_{t=1}^T \mathbb{P}^\pi_{\phi|x_1}\left[\mathcal{Z}^{(1)}_t(x,a;\varepsilon)\right]$$

$$\mathbb{E}^\pi_{\phi|x_1}\left[W^{(2)}_T(x,a;\varepsilon)\right] \le o(\log T) + \sum_{t=1}^T \mathbb{P}^\pi_{\phi|x_1}\left[\mathcal{Z}^{(2)}_t(x,a;\varepsilon)\right],$$

where the events $\mathcal{Z}^{(1)}_t(x,a;\varepsilon)$ and $\mathcal{Z}^{(2)}_t(x,a;\varepsilon)$ are defined as:

$$\mathcal{Z}^{(1)}_t(x,a;\varepsilon) := \left\{(X_t,A_t) = (x,a), \mathcal{E}_t(\varepsilon), \mathcal{E}^{\mathrm{xpr}}_t, (\mathbf{B}^a_{\phi_t} h'_t)(x) \le (\mathbf{B}^{*a}_\phi h^*_\phi)(x) - 2\varepsilon\right\}$$

$$\mathcal{Z}^{(2)}_t(x,a;\varepsilon) := \left\{(X_t,A_t) = (x,a), \mathcal{E}_t(\varepsilon), \mathcal{E}^{\mathrm{xpr}}_t, (\mathbf{B}^a_{\phi_t} h'_t)(x) > (\mathbf{B}^{*a}_\phi h^*_\phi)(x) - 2\varepsilon\right\} \;.$$

The following lemma is proved in Section F.1, and deals events $\mathcal{Z}^{(1)}_t(x,a;\varepsilon)$.

**Lemma S3.** *For structure $\Phi$ with Bernoulli rewards and an ergodic MDP $\phi \in \Phi$, consider $\pi = \mathrm{DEL}(\gamma)$ for $\gamma > 0$. Suppose that (i) $\phi$ is in the interior of $\Phi$; (ii) the solution $\eta^*(\phi)$ is unique for each $(x,a)$ such that $a \notin \mathcal{O}(x;\phi)$; and (iii) continuous at $\phi$. Then, for any $(x,a) \in \mathcal{S} \times \mathcal{A}$ such that $a \notin \mathcal{O}(x;\phi)$,*

$$\lim_{\varepsilon\to 0} \limsup_{T\to\infty} \frac{\sum_{t=1}^T \mathbb{P}^\pi_{\phi|x_1}\left[\mathcal{Z}^{(1)}_t(x,a;\varepsilon)\right]}{\log T} \le (1+\gamma)\eta^*(x,a;\phi) \;.$$

The following lemma is proved in Section F.2, and deals events $\mathcal{Z}_t^{(2)}(x, a; \varepsilon)$. Its proof relies on the following observation. When $\mathcal{Z}_t^{(2)}(x, a; \varepsilon)$ occurs for sufficiently small $\varepsilon$, the facts that $\mathcal{E}_t(\varepsilon)$ holds and that $(\mathbf{B}_{\phi_t}^a h_t')(x) < (\mathbf{B}_\phi^* h_\phi^*)(x) - 2\varepsilon$ imply that $\phi_t(x, a)$ does not estimate $\phi(x, a)$ accurately. The lemma then follows from concentration arguments.

**Lemma S4.** *For structure $\Phi$ with Bernoulli rewards and an ergodic MDP $\phi \in \Phi$, consider $\pi =$ DEL($\gamma$) for $\gamma > 0$. Then, there exists $\varepsilon_2 > 0$ such that for any $(x, a) \in \mathcal{S} \times \mathcal{A}$ such that $a \notin \mathcal{O}(x; \phi)$ and $\varepsilon \in (0, \varepsilon_2)$,*

$$\sum_{t=1}^T \mathbb{P}_{\phi|x_1}^\pi \left[ \mathcal{Z}_t^{(2)}(x, a; \varepsilon) \right] = o(\log T) \quad \text{as } T \to \infty.$$

**B. Upper bound for (S21c).** The last term in (S21) is concerned with the regret generated when $\mathcal{E}_t(\varepsilon)$ does not occur. It is upper bounded in the following lemma proved in Section F.3. To establish this result, we use a similar argument as that in Proposition 5 of [Burnetas and Katehakis, 1997]. Intuitively, we show that by the design of the algorithm, the restricted bias function $h_t'$ is monotonically improved so that it eventually converges to the optimal bias function $h_\phi^*$ with high probability. In this analysis, we provide a more sophisticated concentration inequality than the one in [Burnetas and Katehakis, 1997]. This concentration inequality is particularly important to bound the regret generated in the exploitation phase.

**Lemma S5.** *For structure $\Phi$ with Bernoulli rewards and an ergodic MDP $\phi \in \Phi$, consider $\pi =$ DEL($\gamma$) for $\gamma > 0$. Suppose $\phi$ is in the interior of $\Phi$, i.e., there exists a constant $\zeta_0 > 0$ such that for any $\zeta \in (0, \zeta_0)$, $\psi \in \Phi$ if $\|\phi - \psi\| \le \zeta$. Then, there exists $\varepsilon_3 > 0$ such that for any $\varepsilon \in (0, \varepsilon_3)$,*

$$\mathbb{P}_{\phi|x_1}^\pi[\neg \mathcal{E}_T(\varepsilon)] = o(1/T) \quad \text{as } T \to \infty. \tag{S24}$$

We provide the proof of Lemma S5 in Section F.3. Now, we are ready to complete the proof of Theorem 4. Combining Lemma S3, (S23) and (S22), we get

$$\sum_{x \in \mathcal{S}} \sum_{a \notin \mathcal{O}(x;\phi)} \delta^*(x, a; \phi) \left( \lim_{\varepsilon \to 0} \limsup_{T \to \infty} \frac{\mathbb{E}_{\phi|x_1}^\pi \left[ W_T^{(1)}(x, a; \varepsilon) \right]}{\log T} \right) \le (1 + \gamma) \sum_{x \in \mathcal{S}} \sum_{a \in \mathcal{A}} \delta^*(x, a; \phi) \eta^*(x, a; \phi)$$
$$= (1 + \gamma) K_\Phi(\phi).$$

Similarly, combining Lemma S4 with (S23) and (S22), it follows that for sufficiently small $\varepsilon \in (0, \min\{\varepsilon_2, \varepsilon_3\})$,

$$\limsup_{T \to \infty} \frac{\mathbb{E}_{\phi|x_1}^\pi \left[ \sum_{x \in \mathcal{S}} \sum_{a \notin \mathcal{O}(x;\phi)} SW_T^{(2)}(x, a; \varepsilon) \right]}{\log T} = 0.$$

From Lemma S5, we have that for sufficiently small $\varepsilon \in (0, \min\{\varepsilon_2, \varepsilon_3\})$,

$$\limsup_{T \to \infty} \frac{\mathbb{E}_{\phi|x_1}^\pi \left[ W_T^{(3)}(\varepsilon) \right]}{\log T} = 0.$$

Therefore, recalling the decomposition of regret bound in (S21), we conclude the proof of Theorem 4. □

### F.1 Proof of Lemma S3

To establish the lemma, we investigate the event $\mathcal{Z}_t^{(1)}(x, a; \varepsilon)$ depending on whether $\mathcal{F}_t$ is empty or not, and on whether $\phi_t$ is a good approximation of $\phi$. To this aim, for any given $t > 0$ and $\zeta > 0$, define the event $\mathcal{B}_t(\zeta) := \bigcap_{(x,a) \in \mathcal{S} \times \mathcal{A}} \mathcal{B}_t(x, a; \zeta)$ where for each $(x, a) \in \mathcal{S} \times \mathcal{A}$, $\mathcal{B}_t(x, a; \zeta) := \{\|\phi_t(x, a) - \phi(x, a)\| \le \zeta\}$. Fix $(x, a) \in \mathcal{S} \times \mathcal{A}$ such that $a \notin \mathcal{O}(x; \phi)$. By the continuity assumption

made in Theorem 4, we have:

$$\sum_{t=1}^{T} \mathbb{P}_{\phi|x_1}^{\pi} \left[ \mathcal{Z}_t^{(1)}(x,a;\varepsilon), \mathcal{F}_t \neq \emptyset, \zeta_t < \zeta(\varepsilon), \mathcal{B}_t(\zeta_t) \right]$$

$$\leq \mathbb{E}_{\phi|x_1}^{\pi} \left[ \sum_{t=1}^{T} \mathbb{1}\left[ (X_t, A_t) = (x,a), N_t(x,a) \leq \eta_t(x,a)\gamma_t, \mathcal{F}_t \neq \emptyset, \zeta_t < \zeta(\varepsilon), \mathcal{B}_t(\zeta_t) \right] \right]$$

$$\leq \mathbb{E}_{\phi|x_1}^{\pi} \left[ \sum_{t=1}^{T} \mathbb{1}\left[ (X_t, A_t) = (x,a), N_t(x,a) \leq (\eta^*(x,a;\phi) + \varepsilon)\gamma_t \right] \right]$$

$$\leq (\eta^*(x,a;\phi) + \varepsilon)\gamma_t + 2$$

where the second inequality is from the continuity of $\eta^*(\phi)$, and the last inequality is from a simple counting argument made precise in the following lemma [Burnetas and Katehakis, 1997] (Lemma 3 therein):

**Lemma S6.** *Consider any (random) sequence of $Z_t \in \{0,1\}$ for $t > 0$. Let $N_T := \sum_{t=1}^{T} \mathbb{1}[Z_t = 1]$. Then, for all $N > 0$, $\sum_{t=1}^{T} \mathbb{1}[Z_t = 1, N_t \leq N] \leq N + 1$ (point-wise if the sequence is random).*

**Proof of Lemma S6.** The proof is straightforward from rewriting the summation as follows:

$$\sum_{t=1}^{T} \mathbb{1}[Z_t = 1, N_t \leq N] = \sum_{t=1}^{T} \sum_{n=1}^{\lfloor N \rfloor} \mathbb{1}[Z_t = 1, N_t = n]$$

$$= \sum_{n=1}^{\lfloor N \rfloor} \sum_{t=1}^{T} \mathbb{1}[Z_t = 1, N_t = n] \leq N + 1$$

where the last inequality is from the fact that $\sum_{t=1}^{T} \mathbb{1}[Z_t = 1, N_t = n] \leq 1$. $\qquad\square$

Since $\lim_{T \to \infty} \frac{\gamma_T}{\log T} = (1 + \gamma)$ for all $x \in \mathcal{S}$, we obtain:

$$\lim_{\varepsilon \to 0} \limsup_{T \to \infty} \frac{\mathbb{E}_{\phi|x_1}^{\pi} \left[ \sum_{t=1}^{T} \mathbb{1}\left[ \mathcal{Z}_t^{(1)}(x,a;\varepsilon), \mathcal{F}_t \neq \emptyset, \zeta_t < \zeta(\varepsilon), \mathcal{B}_t(\zeta_t) \right] \right]}{\log T} = (1 + \gamma)\eta^*(x,a;\phi).$$

Hence, to complete the proof of Lemma S3, it suffices to show that

$$\sum_{t=1}^{T} \mathbb{P}_{\phi|x_1}^{\pi} \left[ \mathcal{Z}_t^{(1)}(x,a;\varepsilon), \mathcal{F}_t = \emptyset \right] = O(1) \tag{S25}$$

$$\sum_{t=1}^{T} \mathbb{P}_{\phi|x_1}^{\pi} \left[ \mathcal{Z}_t^{(1)}(x,a;\varepsilon), \mathcal{F}_t \neq \emptyset, \neg\mathcal{B}_t(\zeta_t) \right] = o(\log T) \tag{S26}$$

since $\sum_{t=1}^{T} \mathbb{P}_{\phi|x_1}^{\pi} \left[ \zeta_t > \zeta(\varepsilon) \right] = O(1)$.

To prove (S25), observe that on the event $\mathcal{Z}_t^{(1)}(x,a;\varepsilon)$ for sufficiently large $t \geq e^{e^\varepsilon}$, i.e., $\zeta_t < \varepsilon$, for $b \in \mathcal{O}(x;\phi_t')$, we have

$$\delta^*(x,a;\phi_t, \mathcal{C}_t) = (\mathbf{B}_{\phi_t'}^b h_t')(x) - (\mathbf{B}_{\phi_t}^a h_t')(x)$$

$$\geq (\mathbf{B}_{\phi_t'}^b h_t')(x) - (\mathbf{B}_{\phi}^* h_\phi^*)(x) + 2\varepsilon$$

$$\geq -|(\mathbf{B}_{\phi_t'}^b h_t')(x) - (\mathbf{B}_{\phi}^b h_\phi^*)(x)| + 2\varepsilon$$

$$\geq -(|r_t(x,b) - r_\phi(x,b)| + |h_t'(x) - h_\phi^*(x)|) + 2\varepsilon$$

$$\geq \varepsilon > \zeta_t$$

where the first, second, and fourth inequalities are from that on the event $\mathcal{Z}_t^{(1)}(x,a;\varepsilon)$, $(\mathbf{B}_{\phi_t}^a h_t')(x) \leq (\mathbf{B}_{\phi}^* h_\phi^*)(x) - 2\varepsilon$, $\mathcal{O}(x;\phi_t') \subseteq \mathcal{O}(x;\phi)$, and $|r_t(x,b) - r_\phi(x,b)| + |h_t'(x) - h_\phi^*(x)| \leq \varepsilon$, respectively,

and the last one is from the choice of $t$ such that $\zeta_t = 1/(1 + \log \log t) < \varepsilon$. Therefore, when $\mathcal{Z}_t^{(1)}(x, a; \varepsilon)$ occurs for sufficiently large $t \geq e^{e^\varepsilon}$,

$$\delta_t(x, a) > \zeta_t > 0. \tag{S27}$$

If $\mathcal{F}_t$ is empty, from the design of DEL algorithm, $\delta_t(x, a) > 0$ implies that $\eta_t(x, a) = 0$ and thus $(x, a)$ is not selected in the exploration phase. This concludes the proof of (S25) as $\sum_{t=1}^{T} \mathbb{P}_{\phi|x_1}^{\pi} [\zeta_t > \varepsilon] \leq e^{e^\varepsilon} = O(1)$.

To show (S26), observe that when $\mathcal{Z}_t^{(1)}(x, a; \varepsilon)$ and $\mathcal{F}_t \neq \emptyset$ occur, for $t \geq e^{e^\varepsilon}$ combining (S27) and Lemma S7 given below, we get:

$$\eta_t(x, a) \leq 2SA \left( \frac{S+1}{\zeta_t} \right)^2. \tag{S28}$$

**Lemma S7.** *Consider a structure $\Phi$, an MDP $\phi \in \Phi$, a non-empty correspondence $\mathcal{C} : \mathcal{S} \twoheadrightarrow \mathcal{A}$, and $\zeta > 0$. If $\mathcal{F}_\Phi(\phi; \mathcal{C}, \zeta)$ is non-empty and there exists $(x, a) \in \mathcal{S} \times \mathcal{A}$ such that $\delta^*(x, a; \phi, \mathcal{C}, \zeta) > 0$, then $\eta^*(x, a; \phi, \mathcal{C}, \zeta) \leq 2SA \left( \frac{S+1}{\zeta} \right)^2$ where $\eta^*(x, a; \phi, \mathcal{C}, \zeta)$ is a solution of $\mathcal{P}(\delta^*(\phi, \mathcal{C}, \zeta), \mathcal{F}_\Phi(\phi; \mathcal{C}, \zeta))$.*

**Proof of Lemma S7.** Using the same arguments as those used in Theorem 2 to show that $\mathcal{F}_{\mathrm{un}}(\phi) \subset \mathcal{F}_\Phi(\phi)$, one can easily check that $\mathcal{F}_{\mathrm{un}}(\phi; \mathcal{C}, \zeta) \subset \mathcal{F}_\Phi(\phi; \mathcal{C}, \zeta)$. Note that the diameter of bias function with Bernoulli reward is bounded by $S$. Now for $(x, a) \in \mathcal{S} \times \mathcal{A}$ such that $\delta^*(x, a; \phi, \mathcal{C}, \zeta) > 0$, we have

$$\delta^*(x, a; \phi, \mathcal{C}, \zeta) > \zeta \tag{S29}$$

which then implies that $2 \left( \frac{H_{\phi(\mathcal{C})} + 1}{\delta^*(x, a; \phi, \mathcal{C}, \zeta)} \right)^2 \leq 2 \left( \frac{S+1}{\zeta} \right)^2$. Now let $\eta$ be defined as $\eta(x, a) = \infty$ if $\delta^*(x, a; \phi, \mathcal{C}, \zeta) = 0$ and $\eta(x, a) = 2 \left( \frac{S+1}{\zeta} \right)^2$ otherwise. Then $\eta \in \mathcal{F}_{\mathrm{un}}(\phi; \mathcal{C}, \zeta) \subset \mathcal{F}_\Phi(\phi; \mathcal{C}, \zeta)$. We deduce that the optimal objective value of $\mathcal{P}(\delta^*(\phi, \mathcal{C}, \zeta), \mathcal{F}_\Phi(\phi; \mathcal{C}, \zeta))$ is upper-bounded by

$$\sum_{(x,a) \in \mathcal{S} \times \mathcal{A}} \eta^*(x, a; \phi, \mathcal{C}, \zeta) \delta^*(x, a; \phi, \mathcal{C}, \zeta) \leq \sum_{(x,a) \in \mathcal{S} \times \mathcal{A}} \eta(x, a) \delta^*(x, a; \phi, \mathcal{C}, \zeta)$$

$$\leq 2SA \frac{(S+1)^2}{\zeta}.$$

Using the optimality of $\eta^*(\phi, \mathcal{C}, \zeta)$ and (S29), we conclude that for $(x, a) \in \mathcal{S} \times \mathcal{A}$ such that $\delta^*(x, a; \phi, \mathcal{C}, \zeta) > 0$, $\eta^*(x, a; \phi, \mathcal{C}, \zeta) \leq 2SA \left( \frac{S+1}{\zeta} \right)^2$. $\qquad \square$

From (S28), we deduce by design of DEL that, if $\mathcal{Z}_t^{(1)}(x, a; \varepsilon)$ and $\mathcal{F}_t \neq \emptyset$ occur, for $t \geq e^{e^\varepsilon}$, then:

$$N_t(x, a) \leq \eta_t(x, a) \gamma_t$$

$$\leq 2SA \left( \frac{S+1}{\zeta_t} \right)^2 \gamma_t \leq \gamma_t'$$

where

$$\gamma_t' := 8S^3 A (1 + \gamma)(1 + \log \log t)^2 (\log t + 1) > 2SA \left( \frac{S+1}{\zeta_t} \right)^2 \gamma_t. \tag{S30}$$

Hence defining $\mathcal{B}_t'(x, a) := \{(X_t, A_t) = (x, a), N_t(x, a) \leq \gamma_t', \neg \mathcal{B}_t(\zeta_t)\}$, we get:

$$\sum_{t=1}^{T} \mathbb{P}_{\phi|x_1}^{\pi} \left[ \mathcal{Z}_t^{(1)}(x, a; \varepsilon), \mathcal{F}_t \neq \emptyset, \neg \mathcal{B}_t(\zeta_t) \right] \leq \sum_{t=1}^{T} \mathbb{P}_{\phi|x_1}^{\pi} \left[ \mathcal{Z}_t^{(1)}(x, a; \varepsilon), \mathcal{F}_t \neq \emptyset, \neg \mathcal{B}_t(\zeta_t), t \geq e^{e^\varepsilon} \right] + e^{e^\varepsilon}$$

$$\leq \sum_{t=1}^{T} \mathbb{P}_{\phi|x_1}^{\pi} \left[ \mathcal{B}_t'(x, a) \right] + O(1).$$

Using $\rho > 0$ in (S4), we check that

$$\sum_{t=1}^{T} \mathbb{P}_{\phi|x_1}^{\pi} \left[ \mathcal{B}_t'(x,a) \right]$$

$$\leq \sum_{t=1}^{T} \mathbb{P}_{\phi|x_1}^{\pi} \left[ \min_{y \in \mathcal{S}} N_t(y) \geq \rho t, \mathcal{B}_t'(x,a) \right] + \sum_{t=1}^{T} \mathbb{P}_{\phi|x_1}^{\pi} \left[ \min_{y \in \mathcal{S}} N_t(y) \leq \rho t \right]$$

$$\leq \sum_{t=1}^{T} \mathbb{P}_{\phi|x_1}^{\pi} \left[ \min_{y \in \mathcal{S}} N_t(y) \geq \rho t, \mathcal{B}_t'(x,a) \right] + o(\log T)$$

$$\leq \sum_{t=1}^{T} \mathbb{P}_{\phi|x_1}^{\pi} \left[ \min_{(y,b) \in \mathcal{S} \times \mathcal{A}} N_t(y,b) \geq \frac{\log t}{(1 + \log\log t)^2}, \mathcal{B}_t'(x,a) \right] + o(\log T). \tag{S31}$$

Here, the second inequality stems from (S4) and a union bound (over states). The the last inequality follows from the following lemma:

**Lemma S8.** *Under DEL algorithm, we have*

$$\sum_{t=1}^{T} \mathbb{1} \left[ \min_{y \in \mathcal{S}} N_t(y) \geq \rho t, \min_{(y,b) \in \mathcal{S} \times \mathcal{A}} N_t(y,b) < \frac{\log t}{(1 + \log\log t)^2} \right] = o(\log T). \tag{S32}$$

**Proof of Lemma S8.** For $(x,a) \in \mathcal{S} \times \mathcal{A}$ and $t$ sufficiently large, we claim the following:

$$\mathbb{1} \left[ N_t(x) \geq \rho t, N_t(x,a) < \frac{\log t}{(1 + \log\log t)^2} \right] = 0. \tag{S33}$$

Using the above claim, we can complete the proof. Indeed:

$$\sum_{t=1}^{T} \mathbb{1} \left[ \min_{y \in \mathcal{S}} N_t(y) \geq \rho t, \min_{(y,b) \in \mathcal{S} \times \mathcal{A}} N_t(y,b) < \frac{\log t}{(1 + \log\log t)^2} \right]$$

$$\leq \sum_{t=1}^{T} \sum_{(x,a) \in \mathcal{S} \times \mathcal{A}} \mathbb{1} \left[ \min_{y \in \mathcal{S}} N_t(y) \geq \rho t, N_t(x,a) < \frac{\log t}{(1 + \log\log t)^2} \right]$$

$$\leq \sum_{t=1}^{T} \sum_{(x,a) \in \mathcal{S} \times \mathcal{A}} \mathbb{1} \left[ N_t(x) \geq \rho t, N_t(x,a) < \frac{\log t}{(1 + \log\log t)^2} \right] = O(1) \quad \text{as } T \to \infty.$$

where the first inequality stems from the union bound.

Next we prove the claim (S33). Fix $(x,a) \in \mathcal{S} \times \mathcal{A}$ and consider sufficiently large $t$. Suppose $N_t(x) \geq \rho t$ and let $t_0 = \lfloor \rho t/2 \rfloor$. Then, since $N_{t_0}(x) \leq t_0$, it follows that

$$N_t(x) - N_{t_0}(x) \geq \rho t - \lfloor \rho t/2 \rfloor \geq \rho t/2.$$

Let $t_1 = \min\{u \in \mathbb{N} : u \in [t_0, t], N_u(x) - N_{t_0}(x) = \lfloor \rho t/4 \rfloor\}$ denote the time when the number of visits to state $x$ after time $t_0$ reaches $\lfloor \rho t/4 \rfloor$. Since $N_t(x) - N_{t_0}(x) \geq \rho t/2 \geq \lfloor \rho t/4 \rfloor$, there exists such a $t_1 \in [t_0, t]$. From the construction of $t_1$, it follows that for all $u \in [t_1, t]$, $\lfloor \rho t/4 \rfloor \leq N_u(x)$ and

$$N_t(x) - N_{t_1}(x) = (N_t(x) - N_{t_0}(x)) - (N_{t_1}(x) - N_{t_0}(x))$$
$$\geq \rho t/2 - \lfloor \rho t/4 \rfloor \geq \rho t/4. \tag{S34}$$

Let $\mathcal{N}_{t_1,t}(x) := \{u \in [t_1, t] : X_u = x, \neg \mathcal{E}_u^{\text{mnt}}\}$ be the set of times between $t_1$ and $t$ when the state is $x$ and the algorithm does not enter the monotonization phase and hence checks the condition to enter the estimation phase. For $u \in \mathcal{N}_{t_1,t}(x)$, the condition for the algorithm to enter the estimation phase and select an action with the minimum occurrence is:

$$\exists b \in \mathcal{A} : N_u(x,b) < \frac{\log \lfloor \rho t/4 \rfloor}{1 + \log\log \lfloor \rho t/4 \rfloor} \tag{S35}$$

since from the construction of $t_1$, for any $u \in [t_1, t]$, we have $\frac{\log \lfloor \rho t/4 \rfloor}{1 + \log\log \lfloor \rho t/4 \rfloor} \leq \frac{\log N_u(x)}{1 + \log\log N_u(x)}$.

Now assume that the number of times the algorithm enters the monotonization phase in state $x$ between $t_1$ and $t$ is bounded by $O(\log t)$. From (S34) and (S35), we deduce the desired claim (S33). Indeed, with the observation (S35), the fact that monotonization happens a sublinear number of times implies that the algorithm estimates all actions more than $\frac{\log \lfloor \rho t/4 \rfloor}{1+\log\log\lfloor \rho t/4 \rfloor}$ ($> \frac{\log t}{(1+\log\log t)^2}$) times. Actually, the fact that monotonization happens a sublinear number of times and (S34) imply that $|\mathcal{N}_{t_1,t}(x)| > A\frac{\log\lfloor \rho t/4 \rfloor}{1+\log\log\lfloor \rho t/4 \rfloor}$ for sufficiently large $t$.

Using the following lemma, we bound the number that the algorithm enters the monotonization phase between $t_1$ and $t$:

**Lemma S9.** *For any action $a \in \mathcal{A}$ and three different $u, u', u''$ such that $u < u' < u''$, suppose that the event $\mathcal{E}_t^{\mathrm{mnt}} \cap \{(X_t, A_t) = (x,a)\}$ occurs for all $t \in \{u, u', u''\}$. Then, when $N_u(x) > e$,*

$$N_{u''}(x) - N_u(x) \geq \frac{N_u(x)}{2\log N_u(x)}.$$

**Proof of Lemma S9.** Observe that selecting action $b$ in the monotonization phase at time $t$ means that

$$N_t(x,a) \in [\log^2 N_t(x), \log^2 N_t(x) + 1) \tag{S36}$$

From the fact that $u < u' < u''$, we have $N_{u''}(x,a) \geq N_u(x,a) + 2$ and thus using (S36):

$$\log^2 N_u(x) + 2 \leq N_u(x,a) + 2 \leq N_{u''}(x,a) < \log^2 N_{u''}(x) + 1.$$

We deduce that $\log^2 N_{u''}(x) - \log^2 N_u(x) > 1$, and conclude that for $N_u(x) > e$,

$$N_{u''}(x) - N_u(x) \geq \frac{N_u(x)}{2\log N_u(x)}$$

since the function $\log^2 t$ is concave with derivative $\frac{2\log t}{t}$, i.e., in order to increase $\log^2 t$ by 1, $t$ should be increased by more than $\left(\frac{2\log t}{t}\right)^{-1}$. $\square$

From Lemma S9, it follows that for sufficiently large $t$,

$$\sum_{b\in\mathcal{A}}\sum_{u=t_1}^{t} \mathbb{1}[N_t(x) \geq \rho t, \mathcal{E}_t^{\mathrm{mnt}}, (X_t, A_t) = (x,b)] \leq A\max\left\{3, 3(N_t(x) - N_{t_1}(x))\left(\frac{2\log N_{t_1}(x)}{N_{t_1}(x)}\right)\right\}$$

$$\leq A\max\left\{3, 3(t - \lfloor \rho t/4 \rfloor)\left(\frac{2\log\lfloor \rho t/4 \rfloor}{\lfloor \rho t/4 \rfloor}\right)\right\}$$

$$\leq 24A\log t. \tag{S37}$$

For the first inequality, we apply Lemma S9 with the fact that as $u$ increases, $N_u(x)$ increases and $\frac{2\log N_u(x)}{N_u(x)}$ decreases. The second inequality is from the definition of $t_1$ and (S34). The last inequality holds for sufficiently large $t$. We have completed the proof of Lemma S8. $\square$

We return to the proof of Lemma S3. Lemma S8 establishes (S31). Next we provide an upper bound of (S31). To this aim, we use the following concentration inequality Combes and Proutiere [2014]:

**Lemma S10.** *Consider any $\phi, \pi, \epsilon > 0$ with Bernoulli reward distribution. Define $\mathcal{H}_t$ the $\sigma$-algebra generated by $(Z_s)_{1\leq s\leq t}$. Let $\mathcal{B} \subset \mathbb{N}$ be a (random) set of rounds. Assume that there exists a sequence of (random) sets $(\mathcal{B}(s))_{s\geq 1}$ such that (i) $\mathcal{B} \subset \cup_{s\geq 1}\mathcal{B}(s)$, (ii) for all $s \geq 1$ and all $t \in \mathcal{B}(s)$, $N_t(x,a) \geq \epsilon s$, (iii) $|\mathcal{B}(s)| \leq 1$, and (iv) the event $t \in \mathcal{B}(s)$ is $\mathcal{H}_t$-measurable. Then for all $\zeta > 0$, and $x_1, x, y \in \mathcal{S}, a \in \mathcal{A}$,*

$$\sum_{t\geq 1}\mathbb{P}_{\phi|x_1}^{\pi}[t \in \mathcal{B}, |r_t(x,a) - r_\phi(x,a)| > \zeta] \leq \frac{1}{\epsilon\zeta^2}$$

$$\sum_{t\geq 1}\mathbb{P}_{\phi|x_1}^{\pi}[t \in \mathcal{B}, |p_t(y \mid x,a) - p_\phi(y \mid x,a)| > \zeta] \leq \frac{1}{\epsilon\zeta^2}$$

**Proof of Lemma S10.** Combes and Proutiere [2014] provides a proof of the first part. Now the occurrence of a transition under action $a$ from state $x$ to state $y$ is a Bernoulli random variable, and hence the second part of the lemma directly follows from the first. $\qquad\square$

Let $\mathcal{B}_t''(x,a) := \{\min_{(y,b)\in\mathcal{S}\times\mathcal{A}} N_t(y,b) \geq \frac{\log t}{(1+\log\log t)^2}, \mathcal{B}_t'(x,a)\}$. If at time $t \leq T$, we have the $s$-th occurrence of $\mathcal{B}_t''(x,a)$, then it follows that $s \leq \gamma_t'$ (since $a$ is selected in state $x$ at time $t$, and $N_t(x,a) \leq \gamma_t'$), and thus

$$\min_{(y,b)\in\mathcal{S}\times\mathcal{A}} N_t(y,b) \geq \frac{\log t}{(1+\log\log t)^2} \geq \frac{1}{16S^3 A(1+\gamma)(1+\log\log t)^4}\gamma_t'$$
$$\geq \frac{1}{16S^3 A(1+\gamma)(1+\log\log T)^4}s,$$

where the last inequality follows from $t \leq T$ and $s \leq \gamma_t'$. Thus since $\neg\mathcal{B}_t(\zeta_t)$ holds when $\mathcal{B}_t''(x,a)$ occurs, we deduce that the set of rounds where $\mathcal{B}_t''(x,a)$ occurs satisfies

$$\{t : \mathcal{B}_t''(x,a) \text{ occurs}\} \subset \cup_{s\geq 1} \cup_{(y,b)\in\mathcal{S}\times\mathcal{A}} \{t : s\text{-th occurence of } \mathcal{B}_t''(x,a), N_t(y,b) \geq \epsilon s,$$
$$\|\phi_t(y,b) - \phi(y,b)\| > \zeta_T\},$$

where $\epsilon := \frac{1}{16S^3 A(1+\log\log T)^4}$. Now we apply Lemma S10 to each pair $(y,b)$ with $\zeta = \zeta_T$, and conclude that:

$$\sum_{t=1}^{T} \mathbb{P}_{\phi|x_1}^{\pi}[\mathcal{B}_t''(x,a)] \leq (SA)\frac{16S^3 A(1+\log\log T)^4}{(\zeta_T)^2} = 16S^4 A^2(1+\log\log T)^6 = o(\log T)$$

where the factor $SA$ in the inequality is from the union bound over all $(y,b) \in \mathcal{S}\times\mathcal{A}$. This proves (S26) and completes the proof of Lemma S3. $\qquad\square$

### F.2 Proof of Lemma S4

Let $\varepsilon_2 := \min_{(x,a)\in\mathcal{S}\times\mathcal{A}:a\notin\mathcal{O}(x;\phi)}(\mathbf{B}_\phi^* h_\phi^*)(x) - (\mathbf{B}_\phi^a h_\phi^*)(x) > 0$. Fix $(x,a) \in \mathcal{S}\times\mathcal{A}$ such that $a \notin \mathcal{O}(x;\phi)$, and $\varepsilon \in (0, \varepsilon_2/5)$ so that

$$(\mathbf{B}_\phi^a h_\phi^*)(x) - (\mathbf{B}_\phi^* h_\phi^*)(x) \leq -5\varepsilon. \tag{S38}$$

When $\mathcal{Z}_t^{(2)}(x,a;\varepsilon)$ occurs, we have

$$(\mathbf{B}_{\phi_t}^a h_\phi^*)(x) - (\mathbf{B}_\phi^* h_\phi^*)(x) = (\mathbf{B}_{\phi_t}^a h_\phi^*)(x) - (\mathbf{B}_{\phi_t}^a h_t')(x) + (\mathbf{B}_{\phi_t}^a h_t')(x) - (\mathbf{B}_\phi^* h_\phi^*)(x)$$
$$> (\mathbf{B}_{\phi_t}^a h_\phi^*)(x) - (\mathbf{B}_{\phi_t}^a h_t')(x) - 2\varepsilon$$
$$= \left(\sum_{y\in\mathcal{S}} p_t(y \mid x,a)(h_\phi^*(y) - h_t'(y))\right) - 2\varepsilon$$
$$\geq -3\varepsilon \tag{S39}$$

where the first inequality stems from the fact that $(\mathbf{B}_{\phi_t}^a h_t')(x) > (\mathbf{B}_\phi^* h_\phi^*)(x) - 2\varepsilon$ when $\mathcal{Z}_t^{(2)}(x,a;\varepsilon)$ occurs, and the last inequality follows from the fact that $\mathcal{E}_t(\varepsilon)$ holds when $\mathcal{Z}_t^{(2)}(x,a;\varepsilon)$ occurs.

Let $\zeta = \frac{\varepsilon}{S^2}$. Then, recalling the definition of the event $\mathcal{B}_t(x,a;\zeta) := \{\|\phi_t(x,a) - \phi(x,a)\| \leq \zeta\}$, when $\mathcal{B}_t(x,a;\zeta)$ occurs, we have

$$|(\mathbf{B}_\phi^a h_\phi^*)(x) - (\mathbf{B}_{\phi_t}^a h_\phi^*)(x)| \leq |r_t(x,a) - r_\phi(x,a)| + H_\phi \sum_{y\in\mathcal{S}} |p_t(y \mid x,a) - p_\phi(y \mid x,a)|$$
$$\leq |r_t(x,a) - r_\phi(x,a)| + S^2 \max_{y\in\mathcal{S}} |p_t(y \mid x,a) - p_\phi(y \mid x,a)|$$
$$\leq S^2 \|\phi_t(x,a) - \phi(x,a)\|$$
$$\leq \varepsilon \tag{S40}$$

where for the second inequality, we used $0 \le H_\phi \le S$.

Now, we can deduce that the events $\mathcal{Z}_t^{(2)}(x, a; \varepsilon)$ and $\mathcal{B}_t(x, a; \zeta)$ cannot occur at the same time, i.e.,

$$\mathbb{P}_{\phi|x_1}^\pi \left[ \mathcal{Z}_t^{(2)}(x, a; \varepsilon), \mathcal{B}_t(x, a; \zeta) \right] = 0. \tag{S41}$$

Indeed, when $\mathcal{Z}_t^{(2)}(x, a; \varepsilon) \cap \mathcal{B}_t(x, a; \zeta)$ occurs, (S39) and (S40) imply

$$\begin{aligned}
(\mathbf{B}_\phi^a h_\phi^*)(x) - (\mathbf{B}_\phi^* h_\phi^*)(x) &= (\mathbf{B}_{\phi_t}^a h_\phi^*)(x) - (\mathbf{B}_\phi^* h_\phi^*)(x) - \left( (\mathbf{B}_\phi^a h_\phi^*)(x) - (\mathbf{B}_{\phi_t}^a h_\phi^*)(x) \right) \\
&\ge (\mathbf{B}_{\phi_t}^a h_\phi^*)(x) - (\mathbf{B}_\phi^* h_\phi^*)(x) - |(\mathbf{B}_\phi^a h_\phi^*)(x) - (\mathbf{B}_{\phi_t}^a h_\phi^*)(x)| \\
&\ge -4\varepsilon > -5\varepsilon
\end{aligned}$$

which contradicts (S38) for our choice of $\varepsilon$, i.e., $\varepsilon \in (0, \varepsilon_2/5)$.

Hence, to complete the proof, it is sufficient to show that

$$\sum_{t=1}^T \mathbb{P}_{\phi|x_1}^\pi [(X_t, A_t) = (x, a), \neg \mathcal{B}_t(x, a; \zeta)] = O(1) \tag{S42}$$

as we have the following bound:

$$\begin{aligned}
\sum_{t=1}^T \mathbb{P}_{\phi|x_1}^\pi \left[ \mathcal{Z}_t^{(2)}(x, a; \varepsilon) \right] &= \sum_{t=1}^T \mathbb{P}_{\phi|x_1}^\pi \left[ \mathcal{Z}_t^{(2)}(x, a; \varepsilon), \neg \mathcal{B}_t(x, a; \zeta) \right] \\
&\le \sum_{t=1}^T \mathbb{P}_{\phi|x_1}^\pi [(X_t, A_t) = (x, a), \neg \mathcal{B}_t(x, a; \zeta)]
\end{aligned}$$

where the equality follows from (S41). (S42) is obtained by applying Lemma S10 with $\{(X_t, A_t) = (x, a)\}$, 1 and $\frac{\varepsilon}{S^2}$ for $\mathcal{B}$, $\epsilon$ and $\zeta$, respectively. This complete the proof of Lemma S4.

$\square$

### F.3  Proof of Lemma S5

Recall that:

$$\mathcal{E}_t(\varepsilon) := \left\{ \Pi^*(\phi_t') \subseteq \Pi^*(\phi) \text{ and } |r_t(x, a) - r_\phi(x, a)| + |h_t'(x) - h_\phi^*(x)| \le \varepsilon \, \forall x \in \mathcal{S}, \forall a \in \mathcal{O}(x; \phi_t') \right\}.$$

Hence when $\mathcal{E}_t(\varepsilon)$ occurs, $(i)$ the estimation of the bias function in the restricted MDP $\phi(\mathcal{C}_t)$ is accurate and $(ii)$ the restricted MDP includes the optimal policies of $\phi$. We first focus on the accuracy of the estimated bias function, and then show that the gain of the restricted MDP $\phi(\mathcal{C}_t)$ is monotone increasing and that it eventually includes an optimal policy for the (unrestricted) MDP.

**Estimation error in bias function.** We begin with some useful notations. Let $K := A^S$ be the number of all the possible fixed policies. Fix $\beta \in \left( 0, \frac{1}{K+1} \right)$. For sufficiently large $t > \frac{1}{\frac{1}{K+1} - \beta}$, divide the time interval from 1 to $t$ into $(K+1)$ subintervals $\mathcal{I}_0^t, \mathcal{I}_1^t, \ldots, \mathcal{I}_K^t$ such that $\mathcal{I}_k^t := \{u \in \mathbb{N} : i_k^t \le u < i_{k+1}^t\}$ where $i_0^t := 1$ and $i_k^t := t + 1 - (K + 1 - k)\lfloor \frac{t}{K+1} \rfloor$ for $k \in \{1, ..., K+1\}$. Then, it is easy to check that for each $k \in \{0, ..., K\}$,

$$|\mathcal{I}_k^t| = i_{k+1}^t - i_k^t > \beta t.$$

Indeed, for $k = 0$, $i_1^t - i_0^t = t - K\lfloor \frac{t}{K+1} \rfloor \ge \frac{t}{K+1} > \beta t$, and for $k \in [1, K]$, $i_{k+1}^t - i_k^t = \lfloor \frac{t}{K+1} \rfloor \ge \frac{t}{K+1} > \beta t$ as $t > \frac{1}{\frac{1}{K+1} - \beta}$, i.e., each subinterval length grows linearly with respect to $t$.

For $k \in [0, K]$, $x \in \mathcal{S}$ and $a \in \mathcal{A}$, let $N_k^t(x) := N_{i_{k+1}^t}(x) - N_{i_k^t}(x)$ and $N_k^t(x, a) := N_{i_{k+1}^t}(x, a) - N_{i_k^t}(x, a)$. Using $\rho > 0$ in (S4), for $\zeta > 0$, define an event $\mathcal{D}_t(\zeta)$ as

$$\mathcal{D}_t(\zeta) := \mathcal{D}_t' \cap \mathcal{E}_t'(\zeta) \tag{S43}$$

where we let

$$\begin{aligned}
\mathcal{D}_t' &:= \left\{ N_k^t(x) > \rho \beta t, \forall x \in \mathcal{S}, \forall k \in [0, K] \right\} \\
\mathcal{E}_t'(\zeta) &:= \left\{ \|\phi_u' - \phi(\mathcal{C}_u)\| \le \zeta \, \forall u \in [i_1^t, t] \right\}.
\end{aligned}$$

When $\mathcal{D}_t(\zeta)$ occurs, then in each subinterval, each state is linearly visited, and after the first subinterval, the estimation on the restricted MDP is accurate, i.e., $\phi(\mathcal{C}_t) \simeq \phi_t(\mathcal{C}_t)$. Note that $\mathcal{E}_t(\varepsilon)$ bounds the error in the estimated gain and bias functions. Hence, we establishy the correspondence between $\zeta$ in $\mathcal{D}_t(\zeta)$ and $\varepsilon$ in $\mathcal{E}_t(\varepsilon)$ using the continuity of the gain and bias functions in $\phi$:

**Lemma S11.** *Consider an ergodic MDP $\phi$ with Bernoulli rewards. Then, for $\varepsilon > 0$. there exists $\zeta_0 = \zeta_0(\varepsilon, \phi) > 0$ such that for any $\zeta \in (0, \zeta_0)$, policy $f \in \Pi_D$ and MDP $\psi$, if $\|\psi - \phi\| \leq \zeta$ and $\psi \ll \phi$, then $\psi$ is ergodic, $|g_\psi^f - g_\phi^f| \leq \varepsilon$ and $\|h_\psi^f - h_\phi^f\| \leq \varepsilon$.*

The proof of Lemma S11 is in Section F.3.1. Observe that on the event $\mathcal{D}_t(\zeta)$, for $u \geq i_1^t$, every state is visited more than $\rho\beta t$, i.e., $\log^2 N_u(x) \geq \log^2 \rho\beta t \geq 1$ for all $x \in \mathcal{S}$ and sufficiently large $t$, and thus, for all $(x, a) \in \mathcal{S} \times \mathcal{A}$ such that $a \in \mathcal{C}_u(x)$, $\phi_u(x, a)$ is indeed the estimation of $\phi(x, a)$, i.e., $\phi_u' = \phi_u(\mathcal{C}_u) \ll \phi(\mathcal{C}_u)$. Then, using Lemma S11, it follows that there exists constant $t_0 > 0$ such that for $t > t_0$ and $\zeta \in (0, \min\{\zeta_0(\varepsilon/2, \phi), \varepsilon/2\})$,

$$\mathcal{D}_t(\zeta) \subseteq \{\phi_u' \text{ is ergodic } \forall u \in [i_1^t, t]\} \cap \mathcal{E}_t''(\varepsilon) \tag{S44}$$

where

$$\mathcal{E}_t''(\varepsilon) := \left\{ |r_{\phi_u'}(x, f(x)) - r_\phi(x, f(x))| + |h_{\phi_u'}^f(x) - h_\phi^f(x)| \leq \varepsilon \; \forall u \in [i_1^t, t], \forall f \in \Pi_D(\mathcal{C}_u), \forall x \in \mathcal{S} \right\},$$

and where for restriction $\mathcal{C} : \mathcal{S} \twoheadrightarrow \mathcal{A}$, we denote by $\Pi_D(\mathcal{C})$ the set of all the possible deterministic policies on the restricted MDP $\phi(\mathcal{C})$.

**Monotone improvement.** Based on (S44), we can identify instrumental properties of DEL algorithm when $\mathcal{D}_t(\zeta)$ occurs:

**Lemma S12.** *For structure $\Phi$ with Bernoulli rewards and an ergodic MDP $\phi \in \Phi$, consider $\pi = \text{DEL}$. There exists $\zeta_1 > 0$ and $t_1 > 0$ such that for any $\zeta \in (0, \zeta_1)$ and $t > t_1$, the occurrence of the event $\mathcal{D}_t(\zeta)$ implies that*

$$\Pi^*(\phi_u') \subseteq \Pi^*(\phi(\mathcal{C}_u)), \quad and \quad g_{u+1}^* \geq g_u^*, \quad \forall u \in [i_1^t, t] \tag{S45}$$

*where we denote by $g_u^* := g_{\phi(\mathcal{C}_u)}^*$ and $h_u^* := h_{\phi(\mathcal{C}_u)}^*$ the optimal gain and bias functions, respectively, on the restricted MDP $\phi(\mathcal{C}_u)$ with true parameter $\phi$.*

The proof of Lemma S12 is presented in Section F.3.2.

Define the event

$$\mathcal{M}_t := \left\{ g_{i_{k+1}^t}^* > g_{i_k^t}^* \; \forall k \in [1, K] \text{ or } g_{i_{k+1}^t}^* = g_{i_k^t}^* = g_\phi^* \text{ for some } k \in [1, K] \right\}.$$

Then, by selecting $\zeta$ as in in Lemma S12 and (S44), we can connect the events $\mathcal{M}_t$ and $\mathcal{D}_t(\zeta)$ to the event $\mathcal{E}_t(\varepsilon)$ as follows: for $\zeta \in (0, \min\{\zeta_0(\varepsilon/2, \phi), \varepsilon/2, \zeta_1\})$ and sufficiently large $t > t_1$,

$$\mathcal{M}_t \cap \mathcal{D}_t(\zeta) \subseteq \mathcal{E}_t(\varepsilon). \tag{S46}$$

On the event $\mathcal{M}_t$, there must exists $k \in [1, K + 1]$ such that $g_{i_k^t}^* = g_\phi^*$ since the number $K$ of subintervals is the number of all the possible policy $\Pi^*$. In addition, for such a $k \in [1, K + 1]$, on the event $\mathcal{D}_t(\zeta)$, it follows from Lemma S12 that for all $u \in [i_k^t, t]$, $g_{i_k^t}^* = g_\phi^* \leq g_u^* \leq g_t^*$, i.e., $g_t^* = g_\phi^*$ and thus $\Pi^*(\phi(\mathcal{C}_t)) \subseteq \Pi^*(\phi)$. Therefore, when both of the events $\mathcal{M}_t$ and $\mathcal{D}_t(\zeta)$ occur,

$$\Pi^*(\phi) \supseteq \Pi^*(\phi(\mathcal{C}_t)) \supseteq \Pi^*(\phi_t')$$

(again thanks to Lemma S12). Then, we indeed get $\mathcal{M}_t \cap \mathcal{D}_t(\zeta) \subseteq \mathcal{E}_t(\varepsilon)$: the ergodicity of $\phi_t'$ guaranteed from (S44) implies that the optimal bias function $h_t'$ of $\phi_t'$ is unique, and the event $\mathcal{E}_t''(\varepsilon)$ in (S44) always occurs on the event $\mathcal{D}_t(\zeta)$. Thus the estimated bias function $h_t'$ is close to $h_\phi^*$.

Using (S46) and (S43), for small enough $\zeta \in (0, \min\{\zeta_0(\varepsilon/2, \phi), \varepsilon/2, \zeta_1\})$ and for large enough $t > 0$, we get

$$\mathbb{P}_{\phi|x_1}^\pi [\neg\mathcal{E}_t(\varepsilon)] \leq +\mathbb{P}_{\phi|x_1}^\pi [\neg\mathcal{D}_t(\zeta)] + \mathbb{P}_{\phi|x_1}^\pi [\mathcal{D}_t(\zeta), \neg\mathcal{M}_t]$$
$$\leq O(1) + \mathbb{P}_{\phi|x_1}^\pi [\neg\mathcal{D}_t'] + \mathbb{P}_{\phi|x_1}^\pi [\mathcal{D}_t', \neg\mathcal{E}_t'(\zeta)] + \mathbb{P}_{\phi|x_1}^\pi [\mathcal{D}_t(\zeta), \neg\mathcal{M}_t] \tag{S47}$$

where the first and last inequalities are from (S46) and (S43), respectively. To complete the proof of Lemma S5, we provide upper bounds of each term in the r.h.s. of (S47). the first term can be easily bounded. Indeed, using (S4) and a union bound, we get for $t$ sufficiently,

$$\mathbb{P}^\pi_{\phi|x_1}[\neg\mathcal{D}'_t] \leq \sum_{x\in\mathcal{S}}\sum_{k\in[0,K]}\mathbb{P}^\pi_{\phi|x_1}[N_k^t(x)\leq\rho\beta t] = o(1/t) \qquad\qquad \text{(S48)}$$

where the last equality is from (S4) conditioned on $X_{i_k^t}$ for each $k$.

Lemma S13 below deals with the last term.

**Lemma S13.** *For structure $\Phi$ with Bernoulli rewards and an ergodic MDP $\phi \in \Phi$, consider $\pi = \mathrm{DEL}$. Suppose $\phi$ is in the interior of $\Phi$, i.e., there exists a constant $\zeta_0 > 0$ such that for any $\zeta \in (0,\zeta_0)$, $\psi \in \Phi$ if $\|\phi - \psi\| \leq \zeta$. There exists $\zeta_2 > 0$ such that for $\zeta \in (0,\zeta_2)$,*

$$\mathbb{P}^\pi_{\phi|x_1}[\mathcal{D}_T(\zeta),\neg\mathcal{M}_T] = o(1/T) \quad \text{as } T \to \infty.$$

We provide the proof of Lemma S13 in Section F.3.3. There, the assumption that $\phi$ is in the interior of $\Phi$ plays an important role when studying the behavior of the algorithm in the exploitation phase.

To bound the second term in the r.h.s. of (S47), we use the following concentration inequality:

**Lemma S14.** *Consider any $\pi$ and $x_1 \in \mathcal{S}$. There exist $C_0, c_0, u_0 > 0$ such that for any $(x,a) \in \mathcal{S}\times\mathcal{A}$ and $u \geq u_0$,*

$$\mathbb{P}^\pi_{\phi|x_1}[|\phi_t(x,a) - \phi(x,a)| > \zeta, N_t(x,a) = u] \leq C_0 e^{-c_0 u}.$$

**Proof of Lemma S14.** The proof is immediate from Lemma 4(i) in [Burnetas and Katehakis, 1997], which is an application of Cramer's theorem for estimating Bernoulli random variables. Let $\hat{\phi}_t(x,a)$ be the estimator of $\phi(x,a)$ from $t$ i.i.d. reward and transition samples when action $a$ is selected in state $x$. From Lemma 4(i) in [Burnetas and Katehakis, 1997], there are positive constants $C(x,a)$, $c(x,a)$, and $u_0(x,a)$ (which may depend on $(x,a)$), such that for $u \geq u_0(x,a)$,

$$\mathbb{P}^\pi_{\phi|x_1}[|\phi_t(x,a) - \phi(x,a)| > \zeta, N_t(x,a) = u] \leq \mathbb{P}[|\hat{\phi}_u(x,a) - \phi(x,a)| > \zeta]$$
$$\leq C_0(x,a)e^{-c_0(x,a)u}.$$

We complete the proof by taking $C_0 := \max_{(x,a)\in\mathcal{S}\times\mathcal{A}}C_0(x,a)$, $c_0 := \min_{(x,a)\in\mathcal{S}\times\mathcal{A}}c_0(x,a)$, and $u_0 := \max_{(x,a)\in\mathcal{S}\times\mathcal{A}}u_0(x,a)$. $\qquad\square$

Now observe that:

$$\mathbb{P}^\pi_{\phi|x_1}[\mathcal{D}'_t, \neg\mathcal{E}'_t(\zeta)]$$
$$= \mathbb{P}^\pi_{\phi|x_1}[\mathcal{D}'_t, \|\phi_u(x,a) - \phi(x,a)\| > \zeta, \text{for some } u \in [i_1^t, t], x \in \mathcal{S}, a \in \mathcal{C}_u(x)]$$
$$\leq \sum_{u=i_1^t}^{t}\sum_{x\in\mathcal{S}}\sum_{a\in\mathcal{C}_u(x)}\mathbb{P}^\pi_{\phi|x_1}[\mathcal{D}'_t, \|\phi_u(x,a) - \phi(x,a)\| > \zeta]$$
$$\leq \sum_{u=i_1^t}^{t}\sum_{x\in\mathcal{S}}\sum_{a\in\mathcal{C}_u(x)}\mathbb{P}^\pi_{\phi|x_1}[\|\phi_u(x,a) - \phi(x,a)\| > \zeta, N_u(x,a) \geq \log^2 N_u(x), \rho\beta t \leq N_u(x) \leq u]$$
$$\leq \sum_{u=i_1^t}^{t}\sum_{x\in\mathcal{S}}\sum_{a\in\mathcal{C}_u(x)}\sum_{u'=\rho\beta t}^{u}\sum_{u''=\log^2 u'}^{u'}\mathbb{P}^\pi_{\phi|x_1}[\|\phi_u(x,a) - \phi(x,a)\| > \zeta, N_u(x,a) = u'']$$

where the second inequality follows from the definition of $\mathcal{C}_u(x)$ and the fact that on the event $\mathcal{D}'_t$, $\rho\beta t \leq N_0^t(x) \leq N_u(x)$ for $u \in [i_1^t, t]$. Then, applying Lemma S14, we have

$$\mathbb{P}_{\phi|x_1}^{\pi}[\mathcal{D}'_t, \neg\mathcal{E}'_t(\zeta)] \leq \sum_{u=i_1^t}^{t} \sum_{x\in\mathcal{S}} \sum_{a\in\mathcal{C}_u(x)} \sum_{u'=\rho\beta t}^{u} \sum_{u''=\log^2 u'}^{\infty} C_0 e^{-c_0 u''}$$

$$\leq SAC_0 t^2 \frac{e^{-c_0 \log^2(\rho\beta t)}}{1 - e^{-c_0}}$$

$$= \frac{SAC_0}{1 - e^{-c_0}} t^2 e^{-c_0(\log^2(\rho\beta) + \log t(\log t + 2\log(\rho\beta)))}$$

$$= \frac{SAC_0}{1 - e^{-c_0}} e^{-c_0 \log^2(\rho\beta)} t^{2 - 2c_0 \log(\rho\beta) - c_0 \log t} = o(1/t). \qquad \text{(S49)}$$

Combining Lemma S13, (S48) and (S49) to (S47), we complete the proof of Lemma S5.

$\square$

### F.3.1 Proof of Lemma S11

Define two strictly positive constants:

$$\zeta_r(\phi) := \min\left\{ \frac{r_\phi(x,a)}{2} : \forall x \in \mathcal{S}, \forall a \in \mathcal{A} \text{ s.t. } r_\phi(x,a) > 0 \right\}$$

$$\zeta_p(\phi) := \min\left\{ \frac{p_\phi(y \mid x,a)}{2} : \forall x, y \in \mathcal{S}, \forall a \in \mathcal{A} \text{ s.t. } p_\phi(y \mid x,a) > 0 \right\}.$$

Then, it is straightforward to show that $\phi \ll \psi$ if $\|\psi - \phi\| < \min\{\zeta_r(\phi), \zeta_p(\phi)\}$ since for any $x, y \in \mathcal{S}$ and $a \in \mathcal{A}$, $p_\phi(y \mid x,a) > 0$ implies that $p_\psi(y \mid x,a) \geq p_\phi(y \mid x,a)/2 > 0$, and $r_\phi(x,a) > 0$ implies that $r_\psi(x,a) \geq r_\phi(x,a)/2 > 0$. Therefore, for sufficiently small $\zeta_0 \leq \min\{\zeta_r(\phi), \zeta_p(\phi)\}$, the above observation and the assumption that $\psi \ll \phi$ ensure the mutual absolute continuity between $\phi$ and $\psi$ and thus the ergodicity of $\psi$.

Now, we focus on the continuity of gain and bias functions for given policy $f$. For notational convenience, let $g_\phi^f$ (resp. $g_\psi^f$) and $h_\phi^f$ (resp. $h_\psi^f$) denote the (column) vector of gain and bias functions, respectively, under $\phi$ (resp. $\psi$). Let $P_\phi^f$ (resp. $P_\psi^f$) and $r_\phi^f$ (resp. $r_\psi^f$) are the transition matrix and reward vector w.r.t. policy $f$ under $\phi$ (resp. $\psi$), respectively. Then, we can write the policy evaluation equations of stationary policy $f$ under $\phi$ and $\psi$ as vector and matrix multiplications, c.f., [Puterman, 1994]:

$$g_\phi^f = P_\phi^f g_\phi^f$$
$$h_\phi^f = r_\phi^f - g_\phi^f + P_\phi^f h_\phi^f.$$

Similarly $g_\psi^f = P_\psi^f g_\psi^f$ and $h_\psi^f = r_\psi^f - g_\psi^f + P_\psi^f h_\psi^f$. Since both $\phi$ and $\psi$ are ergodic, by forcing $h_\phi^f(x_1) = h_\psi^f(x_1) = 0$ for some $x_1 \in \mathcal{S}$, the bias functions $h_\phi^f$ and $h_\psi^f$ can be uniquely defined. Let $D^f := P_\phi^f - P_\psi^f$ and $d^f := h_\phi^f - h_\psi^f$. Then, $\|D^f\| \leq S\zeta$ where $\|\cdot\|$ is the max norm. Noting that the ergodicity of $\phi$ and $\psi$ further provides the invertibility of $I - P_\phi^f$ and $I - P_\psi^f$. A basic linear algebra, c.f., Lemma 7 in [Burnetas and Katehakis, 1997], leads to that for any $\varepsilon > 0$, $\|d^f\| \leq \varepsilon$ if

$$\|D^f\| \leq \frac{\varepsilon}{\|(I - P_\phi^f)^{-1}\|(\|h_\phi^f\| + \varepsilon)}$$

where the upper bound is independent of $\psi$. From the above continuity of $h_\psi^f$ (and thus that of $g_\psi^f$) with respect to $\psi$ at $\phi$, we can find $\zeta_0(f, \varepsilon, \phi) > 0$ such that for any $\psi$, $|g_\psi^f - g_\phi^f| \leq \varepsilon$ and $\|h_\psi^f - h_\phi^f\| \leq \varepsilon$ if $\|\psi - \phi\| \leq \zeta_0(f, \varepsilon, \phi) \leq \min\{\zeta_r(\phi), \zeta_p(\phi)\}$. Noting the arbitrary choice of $f \in \Pi_D$, we conclude the proof of Lemma S11 by taking $\zeta_0(\varepsilon, \phi) = \min_{f\in\Pi_D} \zeta_0(f, \varepsilon, \phi)$. $\square$

### F.3.2 Proof of Lemma S12

Let $\varepsilon_1 := \min\{|g_\phi^f - g_\phi^{f'}| : f, f' \in \Pi_D, g_\phi^f \neq g_\phi^{f'}\} > 0$. Let $\zeta_1 := \min\{\zeta_0(\frac{\varepsilon_1}{2S}, \phi), \frac{\varepsilon_1}{2S}\}$ and consider $t$ sufficiently large, i.e., $t > t_0$. For $\zeta \in (0, \zeta_1)$, assume that the event $\mathcal{D}_t(\zeta)$ occurs.

**Proof of the first part of** (S45). Then, for any $u \in [i_1^t, t]$ and $f \in \Pi_D(\mathcal{C}_u)$, it follows from (S44) that for any $x \in \mathcal{S}$,

$$
\begin{aligned}
|g_{\phi'_u}^f - g_\phi^f| &= |(\mathbf{B}_{\phi'_u}^f h_{\phi'_u}^f)(x) - (\mathbf{B}_\phi^f h_\phi^f)(x)| \\
&\leq |r_{\phi'_u}(x, f(x)) - r_\phi(x, f(x))| + \sum_{y \in \mathcal{S}} |h_{\phi'_u}^f(y) - h_\phi^f(y)| \\
&\leq S\frac{\varepsilon_1}{2S} = \frac{\varepsilon_1}{2}
\end{aligned}
\tag{S50}
$$

where the last inequality stems from the definition of $\mathcal{E}_t''(\frac{\varepsilon_1}{2S})$ in (S44). Then, for any $u \in [i_1^t, t]$, $f \in \Pi^*(\phi'_u)$, and $f' \in \Pi_D(\mathcal{C}_u)$, we have:

$$
g_\phi^f \geq g_{\phi'_u}^f - \frac{\varepsilon_1}{2} \geq g_{\phi'_u}^{f'} - \frac{\varepsilon_1}{2} \geq g_\phi^{f'} - \varepsilon_1
$$

where the first and last inequalities stem from (S50), and the second inequality is deduced from the optimality of $f$ under $\phi'_u$. Noting that $f, f' \in \Pi_D(\mathcal{C}_u)$, it follows that $g_{\phi(\mathcal{C}_u)}^f = g_\phi^f \geq g_\phi^{f'} = g_{\phi(\mathcal{C}_u)}^{f'}$. Hence $f$ is optimal under $\phi(\mathcal{C}_u)$ (the choice of $f' \in \Pi_D(\mathcal{C}_u)$ is arbitrary). This completes the proof of the first part in (S45).

**Proof of the second part of** (S45). Fix $u \in [i_1^t, t]$. Assume that

$$
\Pi_D(\mathcal{C}_{u+1}) \cap \Pi^*(\phi'_u) \neq \emptyset.
\tag{S51}
$$

Then, from the first part of (S45), we deduce that:

$$
\Pi_D(\mathcal{C}_{u+1}) \cap \Pi^*(\phi'_u) \subseteq \Pi_D(\mathcal{C}_{u+1}) \cap \Pi^*(\phi(\mathcal{C}_u)).
$$

Combining this with the assumption (S51), we get that $\Pi_D(\mathcal{C}_{u+1}) \cap \Pi^*(\phi(\mathcal{C}_u)) \neq \emptyset$, which implies that $g_{u+1}^* \geq g_u^*$. It remains to prove (S51).

Let $x = X_u$. We first show that:

$$
\mathcal{C}_{u+1}(x) \cap \mathcal{O}(x; \phi'_u) \neq \emptyset.
\tag{S52}
$$

If the algorithm enters the monotonization phase, i.e., the event $\mathcal{E}_u^{\mathrm{mnt}}$ occurs, then it selects action $a = A_u \in \mathcal{C}_u(x) \cap \mathcal{O}(x; \phi'_u)$. We deduce that:

$$
N_u(x, a) \geq \log^2(N_u(x)), \quad N_{u+1}(x, a) = N_u(x, a) + 1, \quad \text{and} \quad N_{u+1}(x) = N_u(x) + 1
$$

Thus, using the fact that $\log^2(n) + 1 > \log^2(n+1)$, we obtain

$$
N_{u+1}(x, a) \geq \log^2(N_u(x)) + 1 \geq \log^2(N_u(x) + 1) = \log^2(N_{u+1}(x)).
\tag{S53}
$$

We have shown that $a \in \mathcal{C}_{u+1}(x)$ and thus $a \in \mathcal{C}_{u+1}(x) \cap \mathcal{O}(x; \phi'_u) \neq \emptyset$.

In case that the event $\mathcal{E}_u^{\mathrm{mnt}}$ does not occur, there must exist an action $a \in \mathcal{O}(x; \phi'_u)$ such that $N_u(x, a) \geq \log^2(N_u(x)) + 1$. Hence, as for (S53), we get:

$$
N_{u+1}(x, a) \geq N_u(x, a) \geq \log^2(N_u(x)) + 1 \geq \log^2(N_u(x) + 1) = \log^2(N_{u+1}(x))
$$

which implies $a \in \mathcal{C}_{u+1}(x) \cap \mathcal{O}(x; \phi'_u) \neq \emptyset$.

Now (S52) implies (S51) (since for any $y \in \mathcal{S}$ such that $y \neq x$, $\mathcal{C}_u(y) = \mathcal{C}_{u+1}(y)$). This completes the proof of the second part in (S45) and that of Lemma S12. $\qquad\square$

### F.3.3 Proof of Lemma S13

We will show that for small enough $\zeta > 0$,

$$
\mathbb{P}_{\phi|x_1}^\pi [\mathcal{D}_t(\zeta), \neg\mathcal{M}_t] = o(1/t)
$$

where we recall
$$\mathcal{M}_t := \left\{ g^*_{i^t_{k+1}} > g^*_{i^t_k} \ \forall k \in [1, K] \text{ or } g^*_{i^t_{k+1}} = g^*_{i^t_k} = g^*_\phi \text{ for some } k \in [1, K] \right\}.$$

For $x \in \mathcal{S}$ and restriction $\mathcal{C} : \mathcal{S} \twoheadrightarrow \mathcal{A}$, define
$$\mathcal{A}^+(x; \phi, \mathcal{C}) := \{ a \in \mathcal{A} : (\mathbf{B}^a_\phi h^*_{\phi(\mathcal{C})})(x) > (\mathbf{B}^*_{\phi(\mathcal{C})} h^*_{\phi(\mathcal{C})})(x) \}$$

as the set of actions that improve the optimal policy of the restricted MDP $\phi(\mathcal{C})$ at state $x$. If $g^*_{\phi(\mathcal{C})} < g^*_\phi$, then there must exist a state $x$ with non-empty $\mathcal{A}^+(x; \phi, \mathcal{C})$. Let $\varepsilon_2 := \min\{ (\mathbf{B}^a_\phi h^f_\phi)(x) - (\mathbf{B}^f_\phi h^f_\phi)(x) : f \in \Pi_D, x \in \mathcal{S}, a \in \mathcal{A}^+(x; \phi, \{f\}) \neq \emptyset \}$. Note that $\varepsilon_2 > 0$.

Define an event
$$\mathcal{M}'_t := \{ \mathcal{E}^{\text{xpt}}_u, \mathcal{A}^+(X_u; \phi, \mathcal{C}_u) \neq \emptyset, \exists u \in [i^t_1, t] \}.$$

Then, we obtain
$$\mathbb{P}^\pi_{\phi|x_1} [\mathcal{D}_t(\zeta), \neg \mathcal{M}_t] \leq \mathbb{P}^\pi_{\phi|x_1} [\mathcal{D}_t(\zeta), \neg \mathcal{M}_t, \neg \mathcal{M}'_t] + \mathbb{P}^\pi_{\phi|x_1} [\mathcal{D}_t(\zeta), \mathcal{M}'_t].$$

We first focus on the last term in the above. Let $\zeta \leq \min\{\zeta_0(\frac{\varepsilon_2}{3S}, \phi), \frac{\varepsilon_2}{3S}, \zeta_0\}$ where $\zeta_0$ is taken from the assumption that $\phi$ is in the interior of $\Phi$, and $t \geq \frac{1}{\beta} e^{e^{\varepsilon_2/3}}$ so that $\zeta_u \leq \varepsilon_2/3$ for any $u \geq i^t_1 \geq \beta t$.

**Bounding $\mathbb{P}^\pi_{\phi|x_1} [\mathcal{D}_t(\zeta), \mathcal{M}'_t]$.** Suppose that for $u \in [i^t_1, t]$ and $x \in \mathcal{S}$, the events $\mathcal{D}_t(\zeta)$ and $\{X_u = x, \mathcal{E}^{\text{xpt}}_u, \mathcal{A}^+(x; \phi, \mathcal{C}_u) \neq \emptyset\}$ occur. From Lemma S12, it directly follows that $\mathcal{O}(x; \phi'_u) \subseteq \mathcal{O}(x; \phi, \mathcal{C}_u)$. By the definition of the improving action set, $\mathcal{O}(x; \phi, \mathcal{C}_u) \cap \mathcal{A}^+(x; \phi, \mathcal{C}_u) = \emptyset$ and thus $\mathcal{O}(x; \phi'_u) \cap \mathcal{A}^+(x; \phi, \mathcal{C}_u) = \emptyset$. Construct $\psi_u$ such that for each $(y, b) \in \mathcal{S} \times \mathcal{A}$,

$$\psi_u(y, b) = \begin{cases} \phi_u(y, b) & \text{if } b \in \mathcal{O}(x; \phi'_u), \\ \phi(y, b) & \text{otherwise}. \end{cases}$$

Note that $\psi_u(\mathcal{C}_u) = \phi'_u$ and thus $\|\psi_u - \phi\| \leq \|\phi'_u - \phi(\mathcal{C}_u)\| \leq \zeta_0$. This implies $\psi_u \in \Phi$ since $\phi$ is an interior point of $\Phi$. For any $a \in \mathcal{A}^+(x; \phi, \mathcal{C}_u) \neq \emptyset$, we get $\delta^*(x, a; \psi_u, \mathcal{C}_u, \zeta_u) = 0$ as:

$$\delta^*(x, a; \psi_u, \mathcal{C}_u) = (\mathbf{B}^*_{\phi'_u} h'_u)(x) - (\mathbf{B}^a_{\psi_u} h'_u)(x) = (\mathbf{B}^*_{\phi'_u} h'_u)(x) - (\mathbf{B}^a_\phi h'_u)(x)$$

$$\leq \frac{2}{3} \varepsilon_2 + (\mathbf{B}^*_{\phi(\mathcal{C}_u)} h^*_{\phi(\mathcal{C}_u)})(x) - (\mathbf{B}^a_\phi h^*_{\phi(\mathcal{C}_u)})(x)$$

$$\leq \frac{2}{3} \varepsilon_2 - \varepsilon_2 = -\frac{1}{3} \varepsilon_2 \leq \zeta_u \qquad \text{(S54)}$$

where the second equality is from the construction of $\psi_u$ and the fact that $\mathcal{O}(x; \phi'_u) \cap \mathcal{A}^+(x; \phi, \mathcal{C}_u) = \emptyset$, i.e., $a \notin \mathcal{O}(x; \phi'_u)$; and the first and second inequalities are from (S44), the definition of $\varepsilon_2$. We have obtained that $\psi_u \in \Phi$ and $\delta^*(x, a; \psi_u, \mathcal{C}_u, \zeta_u) = 0$ for some $a \notin \mathcal{O}(x; \phi'_u)$). Therefore, $\psi_u \in \Delta_\Phi(\phi_u; \mathcal{C}_u, \zeta_u)$. Recalling the entering condition of the exploitation phase, we establish the following relation:

$$\mathcal{D}_t(\zeta) \cap \mathcal{M}'_t \subseteq \left\{ \sum_{x \in \mathcal{S}} \sum_{a \in \mathcal{A}} N_u(x, a) \text{KL}_{\phi_u | \psi_u}(x, a) \geq \gamma_u \ \exists u \in [i^t_1, t] \right\}$$

$$\subseteq \left\{ \sum_{x \in \mathcal{S}} \sum_{a \in \mathcal{A}} N_u(x, a) \text{KL}_{\phi_u | \phi}(x, a) \geq \gamma_u \ \exists u \in [i^t_1, t] \right\}$$

where the last inclusion follows from the construction of $\psi_u$, i.e.,

$$\sum_{x \in \mathcal{S}} \sum_{a \in \mathcal{A}} N_u(x, a) \text{KL}_{\phi_u | \psi_u}(x, a) = \sum_{x \in \mathcal{S}} \sum_{a \notin \mathcal{O}(x; \phi'_u)} N_u(x, a) \text{KL}_{\phi_u | \psi_u}(x, a)$$

$$= \sum_{x \in \mathcal{S}} \sum_{a \notin \mathcal{O}(x; \phi'_u)} N_u(x, a) \text{KL}_{\phi_u | \phi}(x, a)$$

$$\leq \sum_{x \in \mathcal{S}} \sum_{a \in \mathcal{A}} N_u(x, a) \text{KL}_{\phi_u | \phi}(x, a).$$

As a consequence, applying the following lemma, $\mathbb{P}^\pi_{\phi|x_1} [\mathcal{D}_t(\zeta), \mathcal{M}'_t]$ is bounded by $o(1/t)$:

**Lemma S15.** *Consider any $\pi$ and $\phi$ with Bernoulli rewards. Then, for any $\gamma > 0$ and $\rho \in (0,1)$, as $T \to \infty$,*

$$\sum_{t=\rho T}^{T} \mathbb{P}\left[\sum_{x \in \mathcal{S}}\sum_{a \in \mathcal{A}} N_t(x,a)\text{KL}_{\phi_t|\phi}(x,a) \geq (1+\gamma)\log t\right] = o(1/T). \tag{S55}$$

**Proof of Lemma S15.** The proof is an application of Theorem 2 in [Magureanu et al., 2014], which says that for $\gamma' > SA + 1$ and sufficiently large $t > 0$,

$$\mathbb{P}\left[\sum_{x \in \mathcal{S}}\sum_{a \in \mathcal{A}} N_t(x,a)\text{KL}_{\phi_t|\phi}(x,a) \geq \gamma'\right] \leq e^{-\gamma'}\left(\frac{(\gamma')^2 \log t}{SA}\right)^{SA} e^{SA+1}. \tag{S56}$$

Then, putting $(1+\gamma)\log t$ to $\gamma'$, we obtain

$$\sum_{t=\rho T}^{T} \mathbb{P}\left[\sum_{x \in \mathcal{S}}\sum_{a \in \mathcal{A}} N_t(x,a)\text{KL}_{\phi_t|\phi}(x,a) \geq (1+\gamma)\log t\right]$$

$$\leq \sum_{t=\rho T}^{T} e^{-(1+\gamma)\log t}\left(\frac{(1+\gamma)^2(\log t)^3}{SA}\right)^{SA} e^{SA+1}$$

$$\leq \sum_{t=\rho T}^{T} e^{SA+1}\left(\frac{(1+\gamma)^2}{SA}\right)^{SA}\frac{(\log t)^{3SA}}{t^{1+\gamma}}.$$

Using that $(\log t)^{3SA}/t^{1+\gamma} = O(1/t^{1+\gamma/2})$ for $t \geq \rho T$ and $\int_{\rho T}^{T} 1/t^{1+\gamma/2}dt \leq 1/(\rho T)^{1+\gamma/2} = o(1/T)$, we conclude the proof of Lemma S15. $\qquad\square$

**Bounding $\mathbb{P}_{\phi|x_1}^{\pi}[\mathcal{D}_t(\zeta), \neg\mathcal{M}_t, \neg\mathcal{M}_t']$.** From Lemma S12, on the event $\mathcal{D}_t(\zeta)$, it is true that $g_u^*$ is non-decreasing in $u \in [i_1^t, t]$, i.e., $g_{i_k^t}^* \leq g_{i_{k+1}^t}^*$. Hence,

$$\mathcal{D}_t(\zeta) \cap \neg\mathcal{M}_t \subseteq \mathcal{D}_t(\zeta) \cap \left(\cup_{k=1}^{K}\mathcal{M}_t^k\right)$$

where

$$\mathcal{M}_t^k := \{g_{i_k^t}^* = g_{i_{k+1}^t}^* < g_\phi^*\}.$$

Then, it suffices to show that for any $k \in [1, K]$, $\mathbb{P}_{\phi|x_1}^{\pi}[\mathcal{D}_t(\zeta), \mathcal{M}_t^k, \neg\mathcal{M}_t'] = o(1/t)$.

For given $k \in [1, K]$, assume that the event $\{\mathcal{D}_t(\zeta), \mathcal{M}_t^k, \neg\mathcal{M}_t'\}$ occurs. Fix $x \in \mathcal{S}$ such that $\mathcal{A}^+(x; \phi, \mathcal{C}_{i_k^t}) \neq \emptyset$. Since $g_{i_k^t}^* < g_\phi^*$, such a $x \in \mathcal{S}$ must exist. In addition, using the second part of Lemma S12 and recalling (S51) with the fact that the ergodic MDPs $\phi(\mathcal{C}_u), \phi(\mathcal{C}_{u+1})$ have unique bias functions, it follows that $g_u^* = g_{u+1}^*$ and $h_u^* = h_{u+1}^*$ $\forall u \in \mathcal{I}_k^t$. Therefore,

$$\mathcal{A}^+(x; \phi, \mathcal{C}_{i_k^t}) = \mathcal{A}^+(x; \phi, \mathcal{C}_u) \neq \emptyset \; \forall u \in \mathcal{I}_k^t \tag{S57}$$

which implies $\neg\mathcal{E}_u^{\text{xpt}}$ $\forall u \in \mathcal{I}_k^t$ due to the occurrence of $\neg\mathcal{M}_t'$. Recall $N_k^t(x) := N_{i_{k+1}^t}(x) - N_{i_k^t}(x)$ and $N_k^t(x,a) := N_{i_{k+1}^t}(x,a) - N_{i_k^t}(x,a)$. Then, for any $a \in \mathcal{A}$, we can write

$$N_{k+1}^t(x,a) = \sum_{u \in \mathcal{I}_k^t} \mathbb{1}[(X_u, A_u) = (x,a), \neg\mathcal{E}_u^{\text{xpt}}]$$

$$= \sum_{u \in \mathcal{I}_k^t} \mathbb{1}[(X_u, A_u) = (x,a), \mathcal{E}_u^{\text{mnt}} \cup \mathcal{E}_u^{\text{est}}] + \sum_{u \in \mathcal{I}_k^t} \mathbb{1}[(X_u, A_u) = (x,a), \mathcal{E}_u^{\text{xpr}}]$$

$$\leq O(\log t) + \sum_{u \in \mathcal{I}_k^t} \mathbb{1}[(X_u, A_u) = (x,a), \mathcal{E}_u^{\text{xpr}}] \tag{S58}$$

where the last inequality is obtained since by Lemma S9, the number of times the algorithm enters the monotonization phase is $O(\log t)$ (c.f., (S37)), and since by design, the algorithm limits the number of times we enter the estimation phase to $O(\log t/\log\log t)$.

Define $N_k^{t,\mathrm{xpr}}(x,a) := \sum_{u \in \mathcal{I}_k^t} \mathbb{1}[(X_u, A_u) = (x,a), \mathcal{E}_u^{\mathrm{xpr}}]$ and

$$\mathcal{L}_t^k(x,a) := \left\{ N_k^{t,\mathrm{xpr}}(x,a) \geq \frac{\rho\beta t}{2A} \right\}.$$

It is enough to show that for $a \notin \mathcal{A}^+(x; \phi, \mathcal{C}_{i_k^t})$,

$$\mathbb{P}_{\phi|x_1}^\pi \left[ \mathcal{D}_t(\zeta), \mathcal{M}_t^k, \neg\mathcal{M}_t', \mathcal{L}_t^k(x,a) \right] = o(1/t) . \tag{S59}$$

Indeed, it follows from (S58) that on the event $\{ \mathcal{D}_t(\zeta), \mathcal{M}_t^k, \neg\mathcal{M}_t', \neg\mathcal{L}_t^k(x,a) \,\forall a \notin \mathcal{A}^+(x; \phi, \mathcal{C}_{i_k^t}) \}$,

$$\sum_{a \in \mathcal{A}^+(x;\phi,\mathcal{C}_{i_k^t})} N_{k+1}^t(x,a) = N_{k+1}^t(x) - \sum_{a \notin \mathcal{A}^+(x;\phi,\mathcal{C}_{i_k^t})} N_{k+1}^t(x,a)$$

$$\geq \rho\beta t - \sum_{a \notin \mathcal{A}^+(x;\phi,\mathcal{C}_{i_k^t})} N_{k+1}^t(x,a)$$

$$\geq \rho\beta t - \sum_{a \notin \mathcal{A}^+(x;\phi,\mathcal{C}_{i_k^t})} N_k^{t,\mathrm{xpr}}(x,a) - O(\log t)$$

$$\geq \rho\beta t - \frac{\rho\beta t}{2} - O(\log t)$$

which implies that for sufficiently large $t$, there exists $a \in \mathcal{A}^+(x; \phi, \mathcal{C}_{i_k^t})$ such that $N_{k+1}^t(x,a) \geq \frac{1}{3}\rho\beta t \geq \log^2 t \geq \log^2 N_{i_{k+1}^t}(x)$, i.e., $a \in \mathcal{C}_{i_{k+1}^t}$, and thus $g_{i_{k+1}^t}^* > g_{i_k^t}^*$ which contradicts to the occurrence of the event $\mathcal{M}_t^k$, i.e.,

$$\mathbb{P}_{\phi|x_1}^\pi \left[ \mathcal{D}_t(\zeta), \mathcal{M}_t^k, \neg\mathcal{M}_t' \right] \leq \sum_{a \notin \mathcal{A}^+(x;\phi,\mathcal{C}_{i_k^t})} \mathbb{P}_{\phi|x_1}^\pi \left[ \mathcal{D}_t(\zeta), \mathcal{M}_t^k, \neg\mathcal{M}_t', \mathcal{L}_t^k(x,a) \right] .$$

**Bounding $\mathbb{P}_{\phi|x_1}^\pi \left[ \mathcal{D}_t(\zeta), \mathcal{M}_t^k, \neg\mathcal{M}_t', \mathcal{L}_t^k(x,a) \right]$.** It remains to prove (S59). Fix $(x,a) \in \mathcal{S} \times \mathcal{A}$ such that $a \notin \mathcal{A}^+(x; \phi, \mathcal{C}_{i_k^t}) \neq \emptyset$. Assume that the event $\{ \mathcal{D}_t(\zeta), \mathcal{M}_t^k, \neg\mathcal{M}_t', \mathcal{L}_t^k(x,a) \}$ occurs. Let

$$t_3 := \min \left\{ u \in \mathcal{I}_k^t : \sum_{v=i_k^t}^u \mathbb{1}[(X_v, A_v) = (x,a), \mathcal{E}_v^{\mathrm{xpr}}] \geq \frac{\rho\beta t}{4A} \right\}.$$

From the assumption, $t_3 \in \mathcal{I}_k^t$. With a similar argument as that used to derive (S49), we can guarantee $\|\phi_u(x,a) - \phi(x,a)\| \leq \zeta$ for all $u \geq t_3$ with probability $1 - o(1/t)$, i.e.,

$$\mathbb{P}_{\phi|x_1}^\pi \left[ \mathcal{D}_t(\zeta), \mathcal{M}_t^k, \neg\mathcal{M}_t', \mathcal{L}_t^k(x,a), \|\phi_u(x,a) - \phi(x,a)\| > \zeta \,\exists u \in [t_3, i_{k+1}^t] \right]$$

$$\leq \sum_{u=\rho\beta t}^{i_{k+1}^t} \mathbb{P}_{\phi|x_1}^\pi \left[ \|\phi_u(x,a) - \phi(x,a)\| > \zeta, N_u(x,a) \geq \frac{\rho\beta t}{4A} \right]$$

$$\leq \sum_{u=\rho\beta t}^{i_{k+1}^t} C_0 e^{-c_0 \frac{\rho\beta t}{4A}} \leq C_0 t e^{-c_0 \frac{\rho\beta t}{4A}} = o(1/t) \tag{S60}$$

where for the first inequality, we use union bound with the fact that $t_3 \geq i_k^t \geq \rho\beta t$ and $N_u(x,a) \geq \frac{\rho\beta t}{4A}$, and the second inequality is from Lemma S14.

Assume, further, that $\|\phi_u(x,a) - \phi(x,a)\| \leq \zeta$, $\forall u \in [t_3, i_{k+1}^t]$. Then, similarly as in (S54), from the assumption of the correctness of the estimated bias function and (S44), we can deduce that for $u \in [t_3, i_{k+1}^t]$, $\delta^*(x,a; \phi_u, \mathcal{C}_u) > \zeta_u$ and thus $\delta^*(x,a; \phi_u, \mathcal{C}_u, \zeta_u) > \zeta_u$. Hence, in the exploration phase, when $\mathcal{F}_u = \emptyset$, $\eta_u(x,a) = 0$ due to the design of the algorithm, while when $\mathcal{F}_u \neq \emptyset$, $\eta_u(x,a) \leq 2SA \left( \frac{S+1}{\zeta_u} \right)^2$ due to Lemma S7. Therefore, recalling the definition of $\gamma_t'$ in (S30), for any $u \in [t_3, i_{k+1}^t]$, on the event $\{ \mathcal{E}_u^{\mathrm{xpr}}, X_u = x \}$,

$$\eta_u(x,a)\gamma_u \leq 2SA \left( \frac{S+1}{\zeta_u} \right)^2 \gamma_u \leq \gamma_t' = O(\log^2 t)$$

which implies that for sufficiently large $t > 0$ such that $\gamma'_t = O(\log^2 t) < \frac{\rho \beta t}{4A} \leq N_u(x,a) = \Omega(t)$, $\mathbb{1}[\mathcal{E}_u^{\mathrm{xpr}}, (X_u, A_u) = (x,a)] = 0$ for all $u \in [t_3, i_{k+1}^t]$ due to the design of the exploration phase. Hence, it follows that

$$\mathbb{P}^\pi_{\phi|x_1}\left[\mathcal{D}_t(\zeta), \mathcal{M}_t^k, \neg \mathcal{M}_t', \mathcal{L}_t^k(x,a), \|\phi_u(x,a) - \phi(x,a)\| \leq \zeta \; \forall \, u \in [t_3, i_{k+1}^t]\right] = 0.$$

Combining the above with (S60), we have completed the proof of (S59) and thus the proof of Lemma F.3.3.

$\square$