[Reviews · NeurIPS 2018]

Reviewer 1



The paper establishes problem-specific regret lower bounds for ergodic RL problems with arbitrary structure. The paper focuses on MDPs with finite states and actions, and on learning algorithms that are uniformly good over a class of MDPs. The lower bound is made explicit for unstructured MDPs and Lipschitz MDPs. For unstructured MDPs, the bound extends a previous result in [1] to take into account unknown rewards. For Lipschitz MDPs, the bound depends only on the Lipschitz structure, the sub-optimality gap, and the span of the bias functions, and does not explicitly scale with the size of the state space and action space. The paper then proposes an algorithm, DEL, that matches these lower bounds. The proposed lower bounds are a generalization of results in [2] from the bandit setting to the RL setting, and the proposed algorithm is an adaptation of the algorithm proposed in [2]. I think the paper is very clearly written in general. The motivation and intuition for the theorems and proofs are all very well explained. The optimization problem which defines the lower bound still seems a little abstract to me. If possible, it might be helpful to further illustrate the lower bounds through more concrete examples other than Lipschitz MDPs. [1] Apostolos N. Burnetas and Michael N. Katehakis. Optimal adaptive policies for markov decision processes. Mathematics of Operations Research, 22(1):222–255, 1997. [2] Richard Combes, Stefan Magureanu, and Alexandre Proutiere. Minimal exploration in structured stochastic bandits. In Advances in Neural Information Processing Systems 30, 2017.

Reviewer 2



*Summary* This paper deals with undiscounted reinforcement learning. It provides problem-related (asymptotic) lower and upper bounds on the regret, the latter for an algorithm presented in the paper that builds on Burnetas and Katehakis (1997) and a recent bandit paper by Combes et al (NIPS 2017). The setting assumes that an "MDP structure" \Phi (i.e. a set of possible MDP models) is given. The regret bounds (after T steps) are shown to be of the form K_Phi*log T, where the parameter K_\Phi is the solution to a particular optimization problem. It is shown that if \Phi is the set of all MDPs ("the unstructured case") then K_\Phi is bounded by HSA/\delta, where H is the bias span and \delta the minimal action sub-optimality gap. The second particular class that is considered is the Lipschitz structure that considers embeddings of finite MDPs in Euclidian space such that transition probabilities and rewards are Lipschitz. In this case, the regret bounds are shown to not to depend on the size of state and action space anymore. *Evaluation* In my opinion this is an interesting theoretical contribution to RL. Although the analysis is somewhat restricted (ergodic MDPs, asymptotic analysis) and the paper considers only Lipschitz structures, I think the presented results are an important first step in a promising research direction. I did not have time to check the appendix, and the main part does not present any proof sketches either, so I cannot judge correctness. The paper is well-written, if a bit technical. A bit more intuition on how the algorithm works would have been welcome. Overall, in my opinion this is a clear accept. *Comments* - In footnote 4, I think the uniqueness is about the optimal policy, not the bias span. - Algorithm: The role of the quantities s_t(x) and the parameters \epsilon, \gamma should be explained. Also, I wondered whether the minimization in the Exploit step is necessary, or whether an arbitrary element of \mathcal{O} could be chosen as well. An explanation of the difference of the Estimate and Explore steps would be helpful as well. - In l.272 it is claimed that any algorithm for undiscounted RL needs to recompute a new policy at each step. This is not true. Actually, e.g. UCRL and similar algorithms wouldn't work if policies were recomputed at each step. - In the last line "unknown probabilistic rewards" are mentioned as future work, which I did not understand as the current paper (as far as I understood) already deals with such rewards. *Typos and Minor Stuff* - l.112: Footnote 2 is placed unfortunate. - l.142: "state"->"states" - l.154: "algorithms"->"algorithm" - eq.(2b): Delete dot in formula, replace comma by dot. - l.186: "are"->"is" - l.196: "Ee"->"We" - l.203: "makes"->"make" - l.205: Delete "would". - l.225: D might be confused with the diameter. - l.239: "model"->"models" - l.250: "a MDP"->"an MDP" - l.253: "on optimal"->"of the optimal" - l.256: "apparent best"->"apparently best" - l.260: "phase"->"phase)" - l.265: "model"->"model structure" - l.278: "an linear programming"->"a linear programming problem" - l.280: Delete "a". - l.282: "algorithm"->"algorithms" - l.284: "Works"->"Work" - l.300: "An other"->"Another" - l.319: Capitalize "kullback-leibler". PS: Thanks for the detailed comments in the feedback.

Reviewer 3



This paper present an analysis of regret lower bound of structured MDP problems. For Lipschitz MDPs, the regret bound are shown to not grow with the size of states and action spaces. A algorithm called DEL(Direct Exploration Learning) is also proposed to match the regret lower bound. Theorem 1 proves a regret lower bound for arbitrary structured MDP. While the factor K_{\Phi}(\phi) seems a little bit complicated, the authors did a good job explaining it. Theorem 2 handles the special case of unstructured MDP while Theorem 3 deals with Lipschitz structured MDP. The implications of S_{lip} and A_{lip} are obscure without further explanation. The d exponent seems to be indicating that S_{lip} is on the order of S, depending on the other factors in line 234. Without further explanation, that will weaken the analysis on the Lipschitz structure case. Although the proposed algorithm is a proof of concept one, it's satisfactory to see that the lower bound are actually achievable. The results are interesting and the technical details are solid, with that I recommend accept for this paper.